# Understanding Absorption by Black Versus Brown Carbon in Biomass Burning Plumes from the WE-CAN Campaign

Yingjie Shen[1], Rudra P. Pokhrel[1,*], Amy P. Sullivan[2], Ezra J. T. Levin[2,*], Lauren A. Garofalo[3], Delphine K. Farmer[3], Wade Permar[4], Lu Hu[4], Darin W. Toohey[5], Teresa Campos[6], Emily V. Fischer[2], Shane M. Murphy[1]

[1]Department of Atmospheric Science, University of Wyoming, Laramie, WY 82071, USA.
[2]Department of Atmospheric Science, Colorado State University, Fort Collins, CO 80523, USA
[3]Department of Chemistry, Colorado State University, Fort Collins, CO 80523, USA
[4]Department of Chemistry and Biochemistry, University of Montana, Missoula, MT 59812, USA.
[5]Department of Atmospheric and Oceanic Sciences, University of Colorado Boulder, Boulder, CO 80309, USA
[6]National Center for Atmospheric Research, Atmospheric Chemistry Division, Boulder, CO 80301, USA

*now at Air Pollution Control Division, Colorado Department of Public Health and Environment, Denver, CO 80246, USA

*Corresponding author: Shane M. Murphy (shane.murphy@uwyo.edu)*

**Abstract.** Aerosol absorption of visible light has an important impact on global radiative forcing. Wildfires are one of the major sources of light-absorbing aerosol, but there remains significant uncertainty about the magnitude, wavelength dependence and bleaching of absorption from biomass burning aerosol. We collected and analyzed data from 21 Western United States wildfire smoke plumes during the 2018 WE-CAN airborne measurement campaign to determine the contribution of black carbon (BC), brown carbon (BrC), and lensing to the aerosol mass absorption cross-section (MAC). $MAC_{BC}$, MAC of organics ($MAC_{BrC+lensing}$), and the MAC of water-soluble BrC ($MAC_{ws_BrC660}$) are calculated using Photoacoustic Absorption Spectrometer, Single Particle Soot Photometer and Particle-into-Liquid Sampler measurements. $MAC_{BC660}$ does not change significantly with physical age, organic aerosol (OA) concentration, oxygen to carbon ratio (O:C), gas-phase toluene:benzene ratio, modified combustion efficiency (MCE), altitude, or temperature, and has a relatively stable average value of $10.9 \pm 2.1$ $m^2$ $g^{-1}$. On average, 54% of non-BC absorption (23% total absorption) at 660 nm is from water-soluble BrC. $MAC_{ws_BrC660}$ is $0.06 \pm 0.04$ $m^2$ $g^{-1}$ while $MAC_{BrC+lensing}$ is $0.11 \pm 0.06$ $m^2$ $g^{-1}$ at 660 nm, increasing to $0.59 \pm 0.19$ $m^2$ $g^{-1}$ at 405 nm. $MAC_{BrC+lensing}$ is constant with physical age and MCE, but increases slightly with increasing O:C or decreasing toluene:benzene, while total absorption (normalized to CO) slightly decreases with increasing O:C or decreasing toluene:benzene due to decreasing OA. No evidence of BrC bleaching is observed. Comparison to commonly used parameterizations, modeling studies, and the FIREX-AQ observations suggest model overestimation of absorption is likely due to incorrect BrC refractive indices. Quantification of significant brown carbon in the red wavelengths and the stability of $MAC_{BC}$, the observation of minimal bleaching, and the observation of changes in OA with O:C and toluene:benzene markers all serve as important constraints on aerosol absorption in regional and global climate models.

## 1 Introduction

Atmospheric aerosol impact the climate system by directly scattering and absorbing solar radiation, by indirectly changing cloud properties, and through deposition that changes the surface albedo (McConnell et al., 2007; Sarangi et al., 2020). Biomass burning injects a large amount of primary organic aerosol (POA), secondary organic aerosol (SOA) and black carbon (BC) into the atmosphere every year. BC is somewhat poorly defined but is generally considered to be insoluble and refractory and includes a variety of materials such as char, biochar, charcoal, elemental carbon (EC), and soot (Wei et al., 2013). Although it only represents a small fraction of aerosol mass, BC has a significant impact on the global energy budget due to its ability to strongly absorb solar radiation. While still important, positive radiative forcing of BC is lower in IPCC AR6 (2022) than in IPCC AR5 (2013). Bond et al. (2013) estimated the direct radiative forcing for BC from 1750 to 2005 at the top of the atmosphere to be +0.71 W $m^{-2}$, with an uncertainty of 90% while the latest IPCC AR6 (2022) estimates effective radiative forcing for BC from 1750 to 2019 to be +0.11 (-0.2 ~ +0.42) W $m^{-2}$. It is important to note that AR5 reported direct radiative forcing while AR6 reports effective radiative forcing. While BC is emitted from nearly all combustion processes, the largest global source of BC is thought to be biomass burning (Bond et al., 2013). Organic aerosol (OA) also absorbs visible light, but its absorption strongly depends on the wavelength of light (Kirchstetter and Novakov, 2004). Non-BC light absorbing organic compounds are often called brown carbon (BrC) and they are usually co-emitted with BC or formed by secondary chemistry in biomass burning plumes (Andreae and Gelencsér, 2006). Unlike BC, which absorbs light from the UV to the IR, BrC absorption sharply increases in the UV and shorter visible portions of the spectrum and has been historically considered to be almost transparent near the red wavelengths (Andreae and Gelencsér, 2006; Bahadur et al., 2012; Liu et al., 2020). Wildfires in the Western U.S. have increased in recent decades (Westerling et al., 2006; Burke et al., 2021), and will continue increasing according to model predictions (Yue et al., 2013; Hurteau et al., 2014; Ford et al., 2018; Neumann et al., 2021). Therefore, quantitative studies of the radiative effects caused by BC and BrC emitted from wildfires are crucial for a better understanding of future climate and essential to improve climate models.

The large uncertainty in the radiative forcing from BC is caused both by uncertainties in emissions and by uncertainty in properties that affect its optics, such as size distribution, morphology, refractive index, and mixing state (Bond et al., 2006; Kleinman et al., 2020; Brown et al., 2021). For wildfires, most of the aerosol mass is organic (Garofalo et al., 2019). When BC is internally mixed with OA, the BC is coated by other absorbing or non-absorbing materials that cause more photons to interact with the BC core, and therefore enhance the absorption of the BC core. This process is often called the lensing effect even though geometric lensing is not actually happening at these sizes (Fuller et al., 1999). The absorption enhancement caused by the lensing effect is defined as the ratio of the absorption cross-section of a coated BC particle to that of an equivalent uncoated BC particle (Lack and Cappa, 2010). Laboratory experiments have shown a strong absorption enhancement of BC by a factor of two or more (Schnaiter et al., 2003; Schnaiter et al., 2005; Bond and Bergstrom, 2006; Bond et al., 2006; Peng et al., 2016). While observations of absorption enhancement from ambient BC vary widely in field studies due to variations in coating thickness, coating material, source type, or methodological differences, it is often much lower than laboratory values (Liu et al., 2015, 2017; Cappa et al., 2012, 2019; Healy et al., 2015; Krasowsky et al., 2016). Cappa et al. (2019) summarized absorption

enhancements observed at the red end of the visible spectrum from 10 studies including ambient measurements, source sampling, and lab experiments. The absorption enhancement reported by those measurements ranged from 1.1 to 2.8.

Three methods (referred to henceforth as core-shell Mie theory, thermal denuder, and mass absorption cross-section) can be used to obtain estimates of absorption enhancement. Numerical solutions to Mie theory (Bohren and Huffman, 1983) have been used to model aerosol absorption for many years and can be adapted into a core-shell model, which is a simplified version of the complex mixing states in real atmospheric particles (Chylek et al., 2019). The core-shell model assumes particles are concentric spheres, where BC acts as a core and the other materials (typically organics) act as a shell. This model can simulate the absorption enhancement with geometric and optical inputs (i.e., shell thickness, particle radius, refractive index). However, the assumptions made by Mie theory may have significant errors for irregular particles, often found in fresh soot particles and when mixed BC and organics are not concentric spheres. A thermodenuder can be used to remove volatile coating materials by heating them to a temperature from 250 – 400 ºC. The ratio of the absorption coefficient measured in ambient air and measured after passing particles through the thermodenuder gives an empirical absorption enhancement. Liu et al. (2015) utilized a thermodenuder to find the average absorption enhancement at 405 nm and 781 nm to be 1.3 and 1.4, respectively, in the UK during winter. Pokhrel et al. (2017) utilized a photoacoustic absorption spectrometer (PAS) with a thermodenuder and showed that absorption enhancement determined in this manner depends on fuel type and combustion conditions, with absorption enhancements ranging from 0.92 to 1.43 at 660 nm and reaching a maximum of 5.6 at 405 nm. However, a thermodenuder cannot always remove coating materials completely and thus can lead to underestimates of absorption enhancement. The mass absorption cross section of BC ($MAC_{BC}$) is another way to describe the absorbing ability of BC containing particles by describing the absorption per unit mass of BC. $MAC_{BC}$ can be a fundamental input in climate models to convert mass concentration into absorption coefficients (Cho et al., 2019). $MAC_{BC}$ is the particulate absorption divided by the mass of the pure BC at the same wavelength. In this way, the calculated $MAC_{BC}$ will include absorption of the BC core, the absorption and absorption enhancement caused by the coating material. Unfortunately, $MAC_{BC}$ in the ambient atmosphere continues to be not well understood due to the lack of field measurements and instrumental limitations. The atmospheric aging processes on BC can introduce uncertainties on its absorption. Bond and Bergstrom (2006) suggested a $MAC_{BC}$ of 7.5±1.2 $m^2\ g^{-1}$ at 550 nm for fresh BC. Subramanian et al. (2010) reported a $MAC_{BC}$ of 10.9±2.1 $m^2\ g^{-1}$ at 660 nm and 13.1 $m^2\ g^{-1}$ at 550 nm over Mexico City when using a single particle soot photometer (SP2) and the filter-based particle soot absorption photometer (PSAP) instrument during airborne measurements. Krasowsky et al. (2016) reported a $MAC_{BC}$ enhancement of 1.03±0.05 due to the coatings on BC. Zhang et al. (2017) found a $MAC_{BC}$ with a mean of 10 $m^2\ g^{-1}$ and a standard deviation of 4 $m^2\ g^{-1}$ at 660 nm by using both SP2 and PSAP measurements. Cho et al. (2019) summarized $MAC_{BC}$ estimated from more than 10 studies in East and South Asia in both ambient conditions and laboratory experiments, and the values ranged from 4.6 to 11.3 $m^2\ g^{-1}$.

To improve the understanding of the evolution of light-absorbing aerosol from biomass burning, smoke from 21 wildfires in the Western United States were measured near their sources and downwind onboard the NSF/NCAR C-130 aircraft during the Western Wildfire Experiment for Cloud Chemistry, Aerosol Absorption and Nitrogen (WE-CAN) campaign. This campaign represented one of the first airborne attempts to fully characterize Western U.S.

wildfires from several different fuel types, locations, and fire stages (flaming vs. smoldering). This paper presents novel observations about the absorbing properties of the aerosol and compares these observations to modeling studies conducted with the WE-CAN data and to results from the Fire Influence on Regional to Global Environment – Air Quality (FIREX) study conducted in 2019 (Zeng et al., 2021).

## 2 Experimental Method

This work relies on measurements made during the WE-CAN field campaign, which sampled smoke emitted by wildfires across the Western U.S. using the NSF/NCAR C-130 research aircraft. The goal of the campaign was to make detailed observations of the physical, chemical, and optical evolution of aerosol in western wildfire smoke and its impact on climate, air quality, weather, and nutrient cycles. The WE-CAN field campaign consisted of 19 research flights that took place from Jul. 24 – Sep. 13, 2018. Data from 13 flights were analyzed in this study. The flight path and dominant wildfire for each of these flights are shown in Fig. 1.

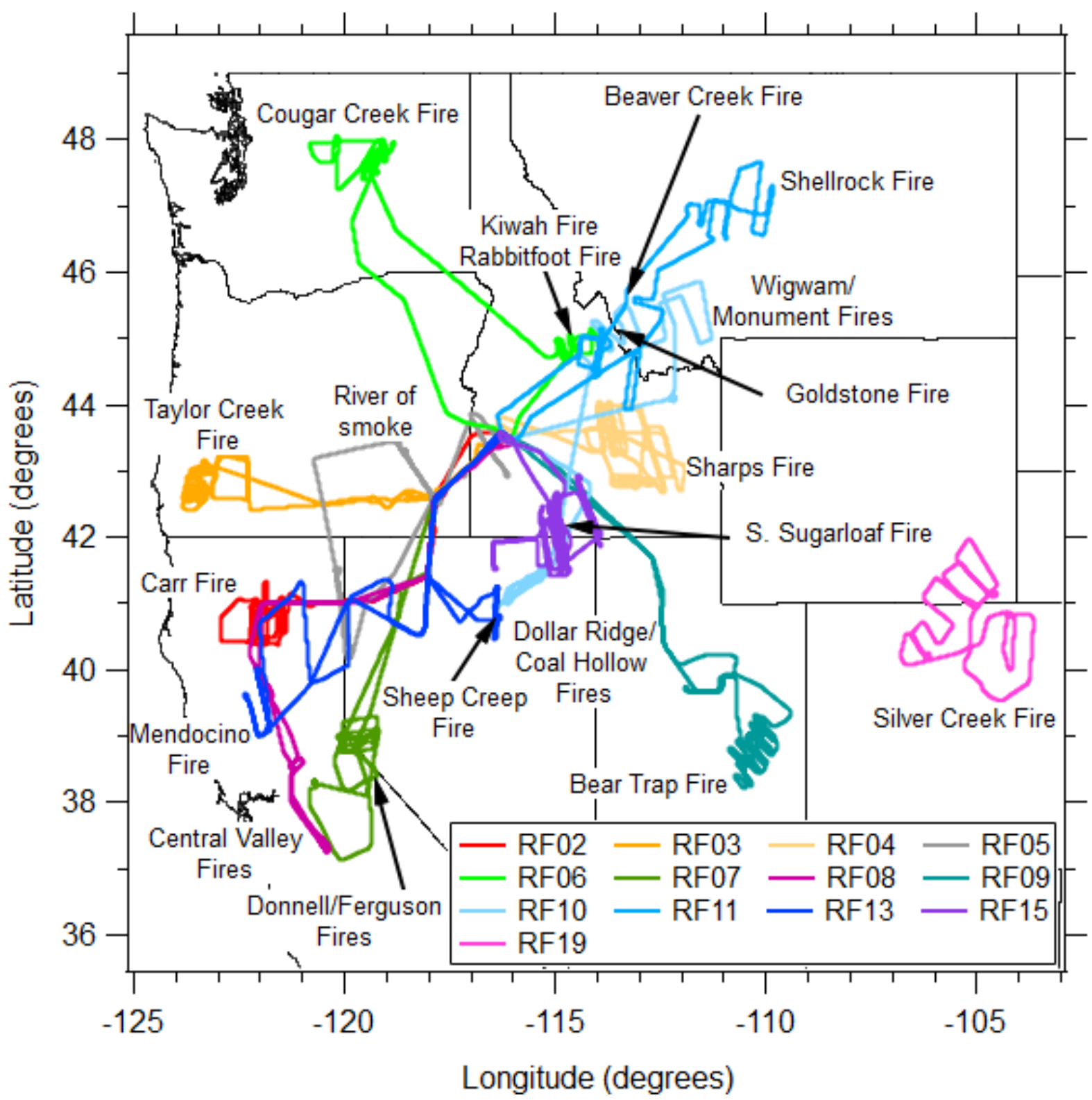

**Figure 1: Flight paths and the sampled wildfires for the WE-CAN flights analyzed in this paper.**

### 2.1 Instrumentation

The following instruments are a subset of those flown during the WE-CAN campaign and are utilized in this work. The full WE-CAN dataset is archived at https://data.eol.ucar.edu/master_lists/generated/we-can. All aerosol instruments utilized in this paper, except the PILS, used pulled air from the same Solid Diffuser Inlet (SDI) inlet. The PILS sampled from a Submicron Aerosol Inlet (SMAI) (Craig et al., 2013a, 2013b, 2014; Moharreri et al., 2014).

#### 2.1.1 Photoacoustic Absorption Spectrometer (PAS)

Aerosol absorption coefficients were measured with the multi-wavelength PAS built by the University of Wyoming (Foster et al., 2019), based on the design of Lack et al. (2012b). A PAS can directly measure the absorption coefficient of dry aerosol. The PAS represents the only way to directly measure aerosol absorption other than the photothermal interferometer (PTI, Sedlacek, 2007), which measures the change in the refractive index of the air near particles caused by absorption. Briefly, when modulated laser light (at the resonant frequency of the cell) is absorbed by the aerosol, pressure waves are created and amplified by the cavity then detected by two microphones (Lack et al., 2006; Foster et al., 2019). The PAS used here has four cells that measure the aerosol absorption coefficient from dry air at 405 nm and 660 nm and thermally denuded air at 405 nm and 660 nm. The denuder was set to 300°C, with the goal of evaporating volatile organic aerosol which might have a potential impact on light absorption. However, absorption from the denuded channels was not used in this study, because the absorption enhancement calculated using the thermodenuder approach is much smaller than the approach taking the ratio of $MAC_{BC}$ to $MAC_{BC\text{-}core}$, and we believe the discrepancy is due to the presence of significant residual organic material after denuding. Two $NO_X$ denuders coated with potassium hydroxide, guaiacol and methanol were installed on the PAS in front of the inlet to remove the absorption from gas-phase $NO_2$ (Williams and Grosjean 1990). A 3 LPM $PM_{2.5}$ cyclone (URG-2000-30ED) was used on the PAS in front of the inlet to provide a $PM_{1.0}$ cut. In addition, a Nafion drier (Purma Pure PD-100T-24MPS) with 100 tubes was installed on the inlet system to dry sample to a relative humidity below 30%. The uncertainty in the absorption coefficient measured by the PAS mainly comes from the calibration technique, in which the highly absorbing substance Regal Black and the CAPS $PM_{SSA}$ were utilized (Foster et al., 2019). The PAS was routinely calibrated (after each flight or every other day if there was a flight everyday) during WE-CAN with an accuracy of +/- 10%.

The PAS microphone shows a pressure-dependent response to pressure. To account for this behavior, we performed pressure-dependent calibration of the PAS where the instrument pressure (both PAS and CAPS $PM_{SSA}$) was dropped stepwise by ~50 torr from ambient to ~300 torr (typical minimum pressure level during WE-CAN). A calibration was performed at each pressure step and the calibration constants were fitted with pressure to get a change in calibration at a desired pressure. Pressure-dependent calibrations were repeated pre and post-campaign to capture variability.

Aerosol optical properties (absorption and extinction) were converted to standard temperature and pressure (STP, 1 atm, 0°C) before data were uploaded. We used temperature and pressure measured by the PAS and CAPS $PM_{SSA}$ to convert optical properties to STP. For absorption, the sample temperature measured by the PAS RH sensor

(Vaisala RH probe) and the pressure measured by the temperature and pressure sensor of the CAPS $PM_{SSA}$ were used. Whereas for extinction, temperature and pressure measured by the CAPS $PM_{SSA}$ were used.

### 2.1.2 Cavity-Attenuated Phase Shift Spectrometer (CAPS $PM_{SSA}$)

After pulling through the $NO_X$ denuder, the $PM_{1.0}$ cyclone, and the Nafion drier in front of the PAS inlet, the sampled air entered through the Aerodyne CAPS $PM_{SSA}$_450 and CAPS $PM_{SSA}$_660 to measure the aerosol scattering and extinction coefficients at 450 nm and 660 nm, respectively. CAPS $PM_{SSA}$ instruments measure extinction by utilizing the cavity attenuated phase shift spectroscopy and measure scattering with an integrating sphere (Onasch et al., 2015). Ammonium sulfate particles were used to calibrate the scattering channel of the CAPS $PM_{SSA}$ during WE-CAN with an accuracy of +/- 3%.

### 2.1.3 Particle-into-Liquid Sampler (PILS) systems

BrC absorption and water-soluble organic carbon (WSOC) were measured by a Particle-into-Liquid Sampler (PILS) system (Sullivan et al., 2022). The PILS continuously collects ambient particles into purified water and provides a liquid sample with the aerosol particles dissolved in it for analysis (Orsini et al., 2003). The size-cut for the PILS was provided by a nonrotating microorifice uniform deposit impactor (MOUDI) with a 50% transmission efficiency of 1 μm (aerodynamic diameter) at 1 atmosphere ambient pressure (Marple et al., 1991). The total airflow for the PILS was approximately 15 LPM. Upstream of the PILS was an activated carbon parallel plate denuder (Eatough et al., 1993) to remove organic gases. In addition, a valve was manually closed periodically for 10 min diverting the airflow through a Teflon filter before entering the PILS allowing for background measurements. The liquid sample obtained from the PILS was pushed through a 0.2 μm PTFE liquid filter by a set of syringe pumps to ensure insoluble particles were removed. The flow was then directed through a liquid waveguide capillary cell (LWCC) and Total Organic Carbon (TOC) Analyzer for near real-time measurement of BrC absorption and WSOC, respectively. More details and a schematic illustration can be found in Zeng et al. (2021).

For the absorption measurement, a 2.5 m path-length LWCC (World Precision Instruments, Sarasota, FL) was used. A dual deuterium and tungsten halogen light source (DH-mini, Ocean Optics, Largo, FL) and absorption spectrometer (FLAME-T-UV-VIS, Ocean Optics, Largo, FL) were coupled to the LWCC via fiber optic cables. Absorption spectra were recorded using the Oceanview Spectroscopy Software over a range from 200 to 800 nm. The wavelength-dependent absorption was calculated following the method outlined in Hecobian et al. (2010). For this study, a 16 s integrated measurement of absorption with a limit of detection (LOD) of 0.1 $Mm^{-1}$ was obtained (Sullivan et al., 2022).

For the WSOC measurement, a Sievers Model M9 Portable TOC Analyzer (Suez Waters Analytical Instruments, Boulder, CO) was used. This analyzer works by converting the organic carbon in the liquid sample to carbon dioxide through chemical oxidation involving ammonium persulfate and ultraviolet light. The carbon dioxide formed was then measured by conductivity. The increase in conductivity observed was proportional to the amount of organic carbon in the liquid sample. The analyzer was run in turbo mode providing a 4 s integrated measurement of WSOC with a LOD of 0.1 μg $C/m^3$ (Sullivan et al., 2022).

### 2.1.4 Single Particle Soot Photometer (SP2)

Refractory black carbon (rBC) number and mass concentrations were measured with a Single Particle Soot Photometer (SP2; Droplet Measurement Technologies) which uses a continuous, 1064 nm Nd:YAG laser to heat absorbing material, primarily rBC, to its vaporization temperature and measures the resulting incandescence (Schwarz et al., 2006). Similar to the CAPS $PM_{SSA}$, the sampled air was sent through the $NO_X$ denuder, $PM_{1.0}$ cyclone, and Nafion drier in front of the PAS inlet before it went to the SP2. The SP2 was calibrated with PSL and size-selected fullerene soot. On the C-130, the SP2 sample line was diluted with HEPA-filtered, pressured ambient air that was passed through a mass flow controller to prevent signal saturation. During post-processing the data was corrected for dilution back to ambient concentrations then to STP.

### 2.1.5 Ultra-High Sensitivity Aerosol Spectrometer (UHSAS)

Particle number concentration was measured by a rack-mounted Ultra-High Sensitivity Aerosol Spectrometer (UHSAS). The flow rate of the rack-mounted UHSAS can be manually lowered by the in-flight operator when the aircraft flew across smoke plumes, so that the UHSAS can stay within its optimum concentration measurement range (Sullivan et al., 2022). The UHSAS was calibrated with ammonium sulfate. The particle mass concentration was calculated by applying these size bins and multiplying by a particle density of 1.4 g $cm^{-3}$ (Sullivan et al., 2022).

### 2.1.6 Proton-Transfer-Reaction Time-of-Flight Mass Spectrometer (PTR-ToF-MS)

The University of Montana proton-transfer-reaction time-of-flight mass spectrometer (PTR-ToF-MS 4000, Ionicon Analytik) was utilized to report the VOC mixing ratios during WE-CAN (Permar et al., 2021). Only the toluene and benzene mixing ratio derived from the PTF-ToF-MS were used in this work; their overall uncertainty is < 15%. More details of the operation, calibration, and validation on the PTR-ToF-MS during WE-CAN can be found in Permar et al. (2021).

### 2.1.7 High-Resolution Aerosol Mass Spectrometry (HR-AMS)

Organic aerosol (OA) was detected by the high-resolution aerosol mass spectrometry (HR-AMS; Aerodyne Inc.). The description of the AMS operation during WE-CAN can be found in Garofalo et al. (2019). The atomic oxygen-to-carbon ratios (O:C) and organic mass-to-organic carbon ratio (OM:OC) used in this work were determined via the improved ambient elemental analysis method for the AMS (Canagaratna et al., 2015). Average (integrated) elemental ratios were obtained by averaging (integrating) elemental masses of carbon, hydrogen, and oxygen and recalculating elemental ratios.

### 2.1.8 Quantum Cascade Laser (QCL) and Picarro Cavity Ring-Down spectrometer (Picarro)

The carbon monoxide (CO) mixing ratio was measured by both an Aerodyne quantum cascade laser instrument (CS-108 miniQCL) and a Picarro cavity ring-down spectrometer (G2401-m WS-CRD) (Garofalo et al., 2019). Because the QCL has better precision than the Picarro instrument, CO measurements from the QCL were preferentially used. However, CO measurements from the Picarro CO data were used for RF10 and RF13, because

the CO-QCL was not operated during those two flights. The carbon dioxide ($CO_2$) mixing ratio was also determined from the Picarro.

### 2.2 Plume Physical Age

The physical age of the plume was calculated by dividing the distance the plume was sampled from the fire source by the in-plume average wind speed. The average wind speed was measured on the NSF/NCAR C-130 aircraft during each plume pass. The distance was estimated by using the longitude and latitude of the geometric center of the plume measured on the NSF/NCAR C-130 and the fire location provided by the U.S. Forest Service. These same plume ages were used by Garofalo et al. (2019), Peng et al. (2020), Lindaas et al. (2021), Permar et al. (2021), and Sullivan et al. (2022).

### 2.3 Plume Integration Method

During the WE-CAN campaign, both the SP2 and PILS had a significant hysteresis compared to other instruments. In the SP2 this is because the sampled air was diluted with particle-free ambient air at various ratios to prevent signal saturation. In the PILS this is because of the retention effect of liquid on the wetted component or within dead volumes (Zeng et al., 2021). Therefore, it was most accurate to integrate properties across airborne transects of wildfire plumes to avoid the impact of instrument hysteresis and measurement noise that can dramatically impact instantaneous ratios. Pseudo-Lagrangian sampling was used during the flights for the WE-CAN campaign, the C-130 aircraft repeatedly crossed the smoke plume from a particular fire by traveling perpendicular to the prevailing winds, crossing the plume, turning, then crossing the plume again further downwind. In this work, we manually identified plume edges based on the inflection point when CO concentrations stopped rapidly changing as we entered and exited the smoke plume. The outside of plume measurement periods had CO mixing ratios from 100 - 300 ppbv. The lowest 10% of each variable from outside plume segments were set to be the background of that variable. If the time between two consecutive outside plume segments was larger than 20 s and the highest CO mixing ratio was 100 ppbv higher than the outside plume CO criteria, this segment was chosen as a plume. The start and end point of each plume was slightly adjusted manually based on the CO mixing ratio to make sure the entire plume was covered. A different start and end point for the SP2 and PILS was adjusted manually based on the rBC mass concentrations and WSOC, respectively.

### 2.4 Absorption Enhancement and Mass Absorption Cross-section

Absorption enhancement ($E_{abs}$) is the ratio of the absorption of the whole particle (including BC core and coating materials) to the absorption of the BC core (Lack and Cappa, 2010). $E_{abs}$ at a specific wavelength ($E_{abs_\lambda}$) was calculated in this study by Eq. 1:

$$E_{abs_\lambda} = \frac{\beta_{Total_\lambda}}{M_{BC} * MAC_{BC_core_\lambda}} \qquad (Eq. 1)$$

where $\beta_{Total_\lambda}$ is the total absorption coefficient at a wavelength of $\lambda$ nm measured by the PAS, $M_{BC}$ is the mass

concentration of BC measured by the SP2, and $MAC_{BC_core_\lambda}$ is the MAC of BC core (without any other coating material) at $\lambda$ nm, which is set to be 6.3 $m^2\ g^{-1}$ at 660 nm (Bond and Bergstrom, 2006; Subramanian et al., 2010).

$MAC_{BC}$ at $\lambda$ nm was calculated following Eq. 2:

$$MAC_{BC_\lambda} = \frac{\beta_{Total_\lambda}}{M_{BC}} \qquad (Eq. 2)$$

$MAC_{BC}$ is utilized more often in this study than $E_{abs}$ because there is not a widely accepted MAC for BC emitted from wildfire. MAC of BrC and lensing is calculated at 405 and 660 nm (Eq. 3):

$$MAC_{BrC+lensing_\lambda} = \frac{\beta_{Total_\lambda} - M_{BC} * MAC_{BC_core_\lambda}}{M_{OA}} \qquad (Eq. 3)$$

where $M_{OA}$ is the organic mass measured by the AMS. Again, the MAC of the BC core is set to be 6.3 and 10.2, respectively, at 660 nm and 405 nm yielding an absorption Ångström exponent (AAE, the negative slope of a logarithmic absorption coefficient against wavelength) of 0.99 for the BC core (Bond and Bergstrom, 2006; Subramanian et al., 2010; Liu, et al., 2015). It should be noted that both BrC and lensing contribute to the $MAC_{BrC+lensing}$, and cannot be separated using this approach.MAC of water-soluble BrC at 660 nm ($MAC_{ws_BrC_660}$) is calculated using Eq. 4:

$$MAC_{ws_BrC_660} = \frac{\beta_{ws_BrC660}}{WSOC} \qquad (Eq. 4)$$

where $\beta_{ws_BrC660}$ is water-soluble light absorption and WSOC is water-soluble organic carbon mass, which are both measured by the PILS system.

To investigate which contributes more to absorption enhancement at 660 nm, the absorption from BrC or the lensing effect, the fractional absorption from BrC at 660 nm is calculated by Eq. 5

$$Fractional\ Abs_{BrC} = \frac{\beta_{BrC_660}}{\beta_{Total_660} - M_{BC} * MAC_{BC_core_660}} \qquad (Eq. 5)$$

where $\beta_{BrC_660}$ is the total BrC absorption coefficient at 660 nm. This is calculated from the water-soluble light absorption provided by the PILS, where we convert absorption from water-soluble BrC to total BrC by WSOM:WSOC and OM:WSOM ratio, and correct absorption from liquid phase to particle phase via Mie theory (more details in 3.1.4, Eq. 9-10). This approach assumes that water insoluble BrC has the same refractive index as water soluble BrC. This assumption would provide a lower estimation on the BrC contribution to the total absorption because Sullivan et al. (2022) found that 45% of the BrC absorption at 405 nm in WE-CAN came from water-soluble species, and Zeng et al. (2022) found that insoluble BrC absorbs more at higher wavelengths than soluble BrC, and methanol-insoluble BrC chromophores caused 87% of the light absorption at 664 nm. $\beta_{Total_660}$ is the total absorption coefficient at 660 nm which is measured by the PAS, $M_{BC}$ is the mass concentration of BC which is measured by the SP2, and $MAC_{BC_core_660}$ is the MAC of the BC core at 660 nm which is set to be 6.3 $m^2\ g^{-1}$ (Bond and Bergstrom, 2006; Subramanian et al., 2010).

### 2.5 Modified Combustion Efficiency (MCE)

The variation of burn condition (e.g., flaming vs. smoldering) and fuel type can cause a significant difference in BC emissions and changes in aerosol properties (Akagi et al., 2011; Andreae, 2019). Burn conditions can be estimated with the modified combustion efficiency (MCE), defined as Eq. 6:

$$MCE = \frac{\Delta CO_2}{\Delta CO + \Delta CO_2} \qquad (Eq. 6)$$

where $\Delta CO_2$ and $\Delta CO$ are the background-subtracted $CO_2$ and $CO$ mixing ratio. The background of $CO_2$ and $CO$ mixing ratio is obtained via the same process described in Section 2.3.

### 2.6 Absorption, Scattering and Single Scattering Albedo (SSA)

Plume integrated absorption and scattering were normalized ($x/CO$) by taking the ratio of background-subtracted absorption or scattering ($\Delta x$) to the background-subtracted $CO$ mixing ratio ($\Delta CO$) (Eq. 7), so that the changing of the normalized properties is not impacted by dilution of the plume with background air.

$$x/CO = \frac{\Delta x}{\Delta CO} \qquad (Eq. 7)$$

Although the CAPS $PM_{SSA}$ provides scattering directly, the scattering is only accurate if extinction is below 1000 $Mm^{-1}$ (Onasch et al., 2015). We calculated scattering by subtracting absorption measured by the PAS from extinction measured by the CAPS $PM_{SSA}$ to avoid high uncertainty caused by extremely dark plumes. Similarly, SSA at a specific wavelength ($\lambda$) was also calculated by using both the PAS absorption ($\beta_{Total_\lambda}$) and CAPS $PM_{SSA}$ extinction ($Ext_\lambda$) (Eq. 8).

$$SSA_\lambda = \frac{Ext_\lambda - \beta_{Total_\lambda}}{Ext_\lambda} \qquad (Eq. 8)$$

## 3 Results and Discussion

### 3.1 Absorption of BC and BrC at Red Wavelengths

#### 3.1.1 Mass Absorption Cross-Section of Black Carbon at 660 nm

Plume integrated MCE, $MAC_{BC}$ at 660 nm ($MAC_{BC660}$) and BC:OA ratio from the 13 WE-CAN research flights with clear plume transects of biomass burning plumes are shown in Fig. 2. Even fire plumes from individually named fires are usually a mix of many different burning conditions, and it is hard to identify the exact source in most wildfire smoke measurements, especially for well mixed plumes. Therefore flight-to-flight data is analyzed in this study because each flight covered a region, and an overall behavior of absorbing aerosol from wildfire can be provided. $MAC_{BC660}$ varies between different flights with RF03 having the highest average $MAC_{BC660}$ of 12.9 $m^2$ $g^{-1}$, (median MCE of 0.94, median BC:OA of 0.015) and RF10 having the lowest average $MAC_{BC660}$ of 8.6 $m^2$ $g^{-1}$, (median MCE of 0.88, , median BC:OA of 0.011). The average of all plume-integrated $MAC_{BC660}$ is 10.9 $m^2$ $g^{-1}$, with a standard deviation of 2.1 $m^2$ $g^{-1}$. This result is similar to some other recent airborne measurements. Subramanian et al. (2010) reported a $MAC_{BC660}$ of 10.9 ± 2.1 $m^2$ $g^{-1}$ using a SP2 and PSAP operated during the MILAGRO campaign, which

included airborne measurements for biomass fires over Mexico. Similarly, Zhang et al. (2017) estimated a $MAC_{BC660}$ of 10 $m^2$ $g^{-1}$ utilizing both SP2 and PSAP deployed on the NASA DC-8 research aircraft for the DC3 campaign, which measured the upper tropospheric BC over the central U.S. Taylor et al. (2020) calculated a $MAC_{BC655}$ of 12 ± 2 $m^2g^{-1}$ over the southeast Atlantic Ocean, using airborne measurements from a SP2 and PAS in the CLARIFY-2017 campaign.

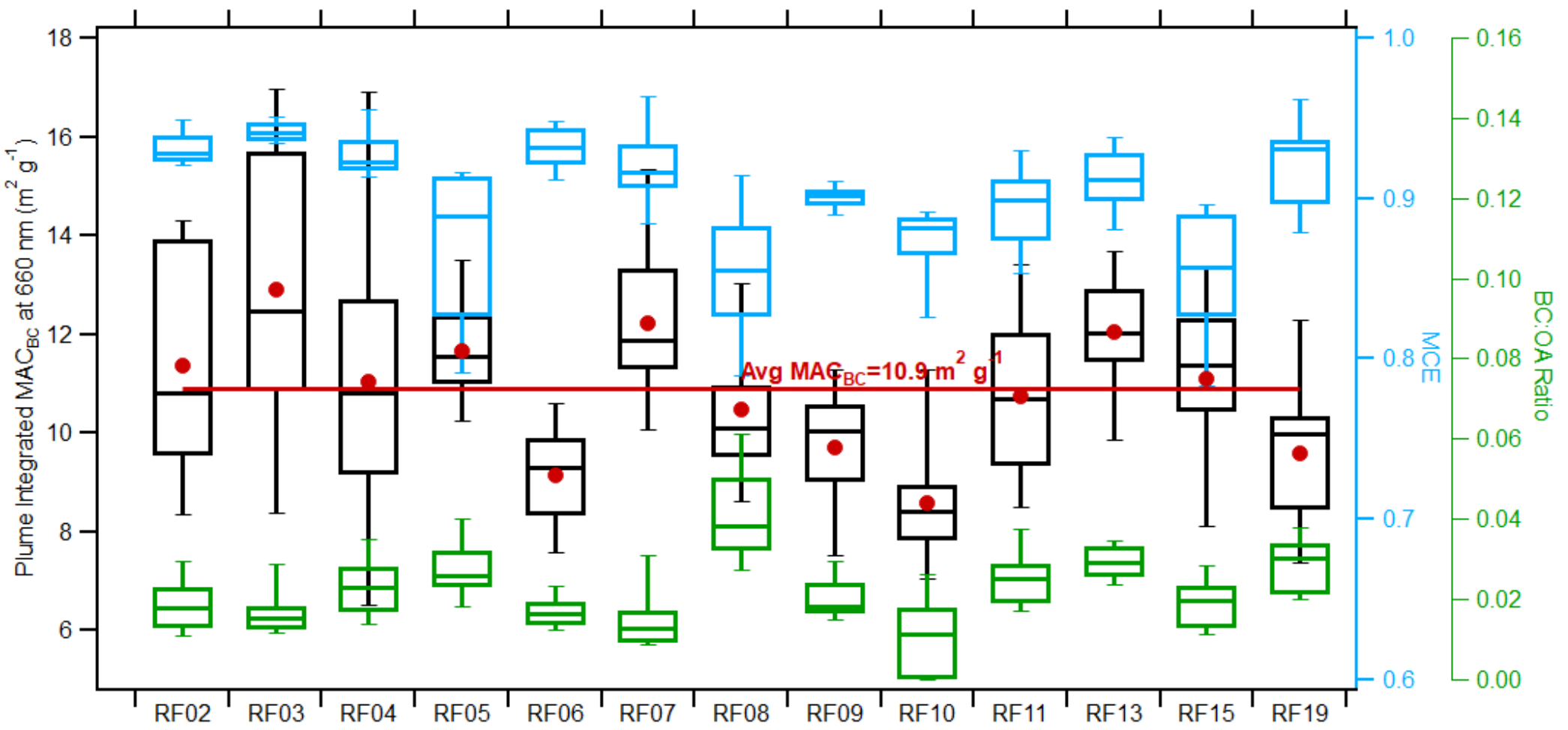

**Figure 2: Box plots of plume integrated MCE (blue box), $MAC_{BC660}$ (black box) and BC:OA (green box) for each flight. On each box the central line represents the median, the top and bottom edge represents 75% and 25%, the top and bottom whiskers represent 90% and 10%, and the red dot shows the average. The red line indicates the average value for all plume integrated $MAC_{BC660}$.**

These results are encouragingly similar given the breadth of measurement techniques (PSAP is filter-based whereas PAS is a direct measurement), geographic regions (Continental U.S. for DC3, Mexico for MILAGRO, African outflow for CLARIFY) and altitude in the atmosphere (all were airborne campaigns covering a range of altitudes). If we apply 6.3 $m^2$ $g^{-1}$ as the MAC of a BC core at 660 nm (Bond and Bergstrom, 2006; Subramanian et al., 2010), then the average absorption enhancement for the entire campaign is 1.7. This means the absorption of coated BC is 1.7 times higher than bare BC at 660 nm, which is close to the factor of ~2 reported by laboratory experiments (Schnaiter et al., 2005; Peng et al., 2016), larger than some field measurements (Cappa et al., 2012&2019; Healy et al., 2015), but close to 1.85 ± 0.45 measured by Taylor et al. (2020) in African biomass burning plumes. The similarity to the Taylor et al. (2020) result suggests global similarities in the $MAC_{BC660}$ from aerosol emitted from wildfires.

The variation of MCE in different flights is caused by the different fuel sources and burning characteristics of the measured fires. Fires with high MCE tend to have more flaming combustion while those with lower MCE tend to have more smoldering combustion. Because of this, fires with different MCE may produce different coating material and thus changing $MAC_{BC660}$. MCE can vary in the same flight (Fig. 2), for example RF05 (mixture of multiple fire sources) and RF15 (single fire sources), because multiple fires were measured in some flights. Several other factors, such as physical age and chemical age (discussed later) also may impact $MAC_{BC660}$, but we first investigate if there was a clear relationship between MCE and $MAC_{BC660}$.

The comparison between plume integrated $MAC_{BC660}$ and MCE is shown in Fig. S1. No clear relationship between $MAC_{BC660}$ and MCE can be seen from individual flights or amongst all the flights combined ($R^2$=0.01). This result indicates that the combustion conditions (flaming or smoldering) does not have an easily described relationship to $MAC_{BC660}$. This poor relation is similar to the relationship observed by Pokhrel et al. (2016). This lack of relationship is likely due to the difficulty for MCE to predict aerosol properties such as BC:OA (Grieshop et al., 2009), upon which effective organic aerosol absorptivity highly depends (Saleh et al., 2014).

### 3.1.2 Relationship of Bulk Optical Properties at 660 nm to Physical Age

Figure 3 shows the evolution of $MAC_{BC660}$, SSA, scattering and absorption at 660 nm versus the time since emission, which will be referred to from now on as physical age. While some flights only intercepted fresh plumes (for example, RF03 and RF15, with a physical age less than 200 min), others intercepted relatively old plumes (for example, RF02 and RF11, with a physical age of 600 min and 800 min, respectively). To eliminate the influence of dilution of particles with time, scattering and absorption are normalized by taking the ratio of the enhancement of scattering or absorption above the background to the enhancement of CO concentration above the background. CO is a conserved tracer that does not react on timescales relevant to these observations. No clear trend between $MAC_{BC660}$ and physical age is apparent from individual flights (Fig. 3a), linear fitting of all flights combined (black solid line) gives a slope close to zero demonstrating that the absorption enhancement changes little with physical age, which is consistent with the results from Subramanian et al. (2010).

However, SSA at 660 nm ($SSA_{660}$) shows a slightly increasing trend with physical age (Fig. 3b) increasing from 0.94 to 0.97 in 10 hours, though the correlation is not very strong with a $R^2$ of 0.14. The increase of $SSA_{660}$ is partly caused by the increase of scattering at 660 nm (Fig. 3c). The particle size increases with age (Fig. S2) due to coagulation of small particles and condensation of vapors. The volume mean diameter of the particles increased from on average 0.18 μm to 0.34 μm across all the plumes detected. Even for each individual flight, the increasing trend in particle mean diameter is clear. Another contributor to increasing SSA is the decrease in absorption at 660 nm (Fig. 3d) with age for most fires. Overall, the trends in SSA, absorption, and scattering with physical age are quite weak with poor correlation coefficients.

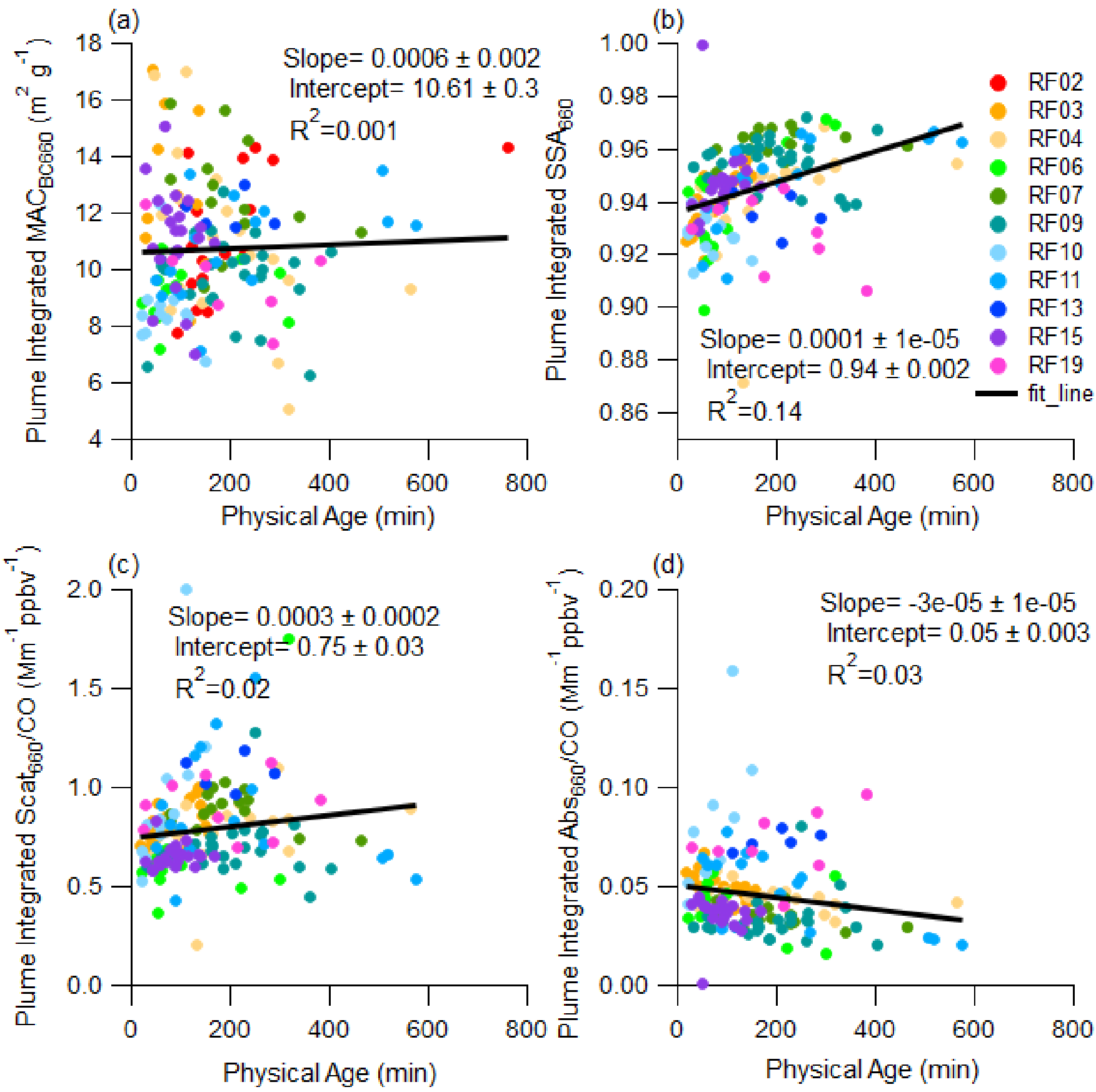

**Figure 3: Time evolution of plume integrated (a) $MAC_{BC660}$, (b) $SSA_{660}$, (c) scattering, and (d) absorption at 660 nm.**

### 3.1.3 Response of Optical Properties at 660 nm to Markers of Chemical Oxidation

$MAC_{BC660}$ was next compared to chemical markers of oxidation, since physical age did not provide strong correlations and often does not do a good job of representing oxidation and photochemical reactions that occur in plumes. Organic coatings of BC cores may be removed via these reactions, or new species may condense on the BC core during chemical reactions, both of which would further change the optical properties of BC. Figure 4 shows the evolution of plume integrated $MAC_{BC660}$ versus chemical clocks based on (a) the ratio of gas-phase toluene:benzene and (b) the particle-phase oxygen-to-carbon (O:C) ratio. The toluene:benzene ratio decreases with photochemical processing time since toluene is more reactive than benzene (Gouw et al., 2005), while the O:C ratio characterizes the oxidation state of OA and typically increases with photochemical age (Aiken et al., 2008).

As shown in Fig. 4 the toluene:benzene ratio ranges from 0.33 to 0.88 across all flights while the O:C ratio is between 0.35 and 0.72. It is difficult to discern any pattern of $MAC_{BC660}$ changing with either marker in an individual flight. However, the trends are slightly clearer after combining plumes from all the flights. From the linear fit line

(black solid line) to data from all flights, the negative slope of -5.7 between $MAC_{BC660}$ and toluene:benzene ratio (Fig. 4a) infers that $MAC_{BC660}$ is larger when toluene:benzene is lower, which is typically thought to indicate more extensive oxidation has occurred. The positive slope of 2.2 between $MAC_{BC660}$ and O:C ratio (Fig. 4b), supports the idea of larger $MAC_{BC660}$ with more oxidation. However, correlations are poor ($R^2 < 0.2$) and because these trends are not visible within a single plume, the explanation for the trends must be either that different fires emit different O:C and toluene:benzene ratios, or that the chemistry that created the observed ratios occurred before the first transect of a plume. Figure S3 supports this explanation in that while there is chemical aging within flights, the O:C and toluene:benzene ratios are more variable from flight to flight. For example, RF06 got more chemical aged with time, but the chemical markers for RF13 were flat with time. For either mechanism, the data shows that plumes that appear "older" either by photochemical aging or because of more aged appearing emissions have a slightly higher $MAC_{BC660,}$ though the main point is that the $MAC_{BC660}$ does not change dramatically with either physical or chemical age for the observations during WE-CAN.

Figure 4c and 4d show that the plume-integrated mass concentration of BC ($M_{BC}$, from the SP2) normalized by CO (the ratio of background subtracted $M_{BC}$ to background subtracted CO) decreases with the toluene:benzene and O:C ratios. One would expect a constant value of $M_{BC}$/CO for single plumes from an individual fire because both are primary and inert. Indeed, there is no obvious decreasing of $M_{BC}$ ($R^2$ <0.5) within an individual flights except for RF13. The decreasing of $M_{BC}$ with markers of chemical age amongst all the flights appears to be due to the different properties of the different fires near the source. Different fires tend to have different O:C ratio and toluene:benzene ratios, as shown in Fig. S4. Therefore, the decreasing trend of $M_{BC}$ with markers of chemical age is more likely caused by fire-to-fire properties or rapid aging at the source rather than aging of the plume after the initial transect.

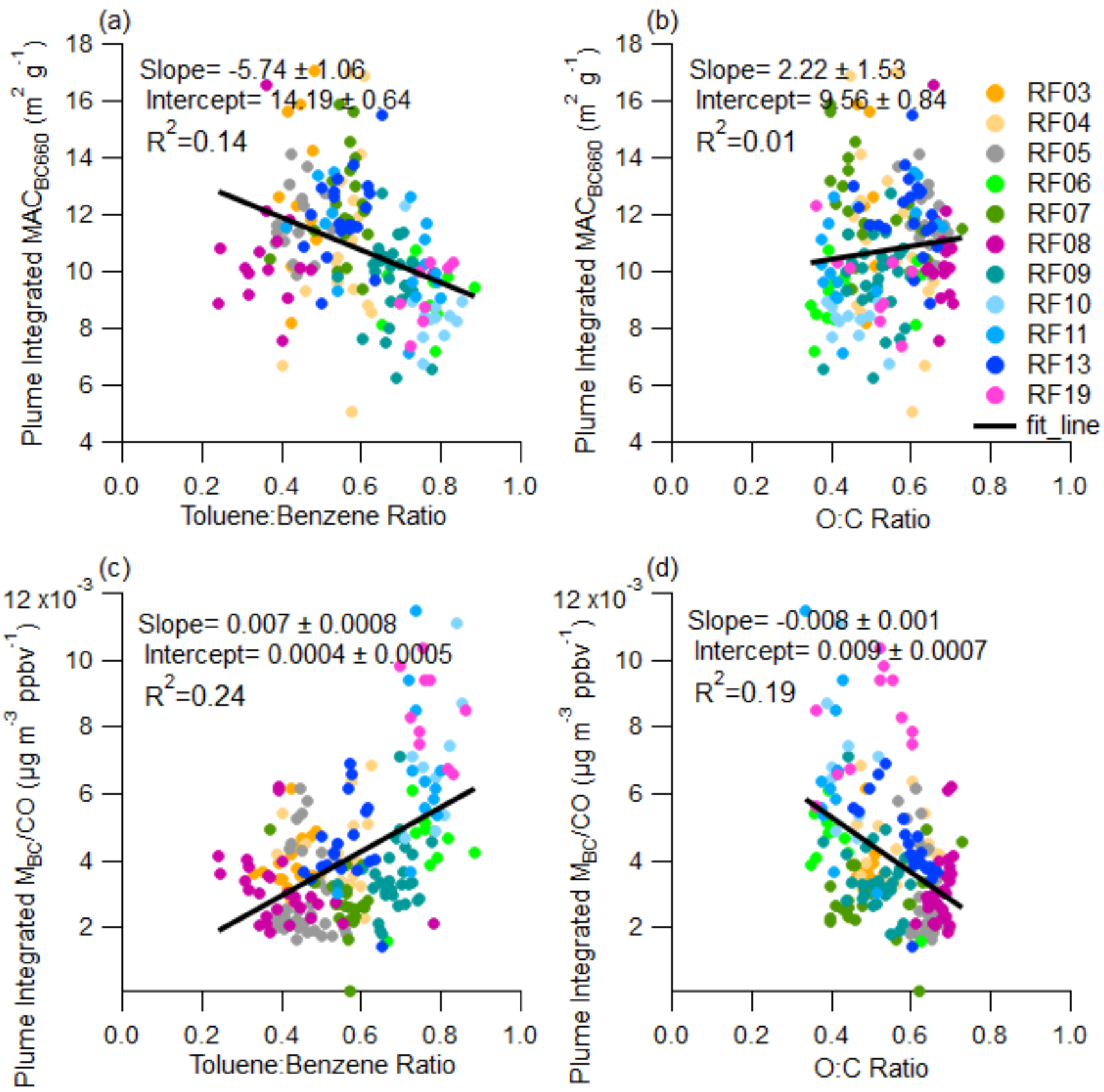

**Figure 4: Plume integrated $MAC_{BC660}$ variations with (a) toluene:benzene ratio and (b)O:C ratio; plume integrated $M_{BC}$ variations with (c) toluene:benzene ratio and (d)O:C ratio.**

SSA, scattering, and absorption at 660 nm are compared with the toluene:benzene ratio and O:C ratio in Fig. 5. There is a slight trend of increasing $SSA_{660}$ with these ratios that would correspond to more oxidized aerosol, but the correlation is poor ($R^2$=0.04). The increase of $SSA_{660}$ with markers of chemical age is consistent with the work from Kleinman et al. (2020) on Western U.S. wildfire emissions, although $-Log_{10}(NO_x/NO_y)$ was used as the indicator for photochemical age in their study.

Both scattering and absorption at 660 nm decrease with ratios corresponding to more chemical aging, which suggests that the amount of absorbing material has changed. The comparison between normalized OA and these markers of chemical age demonstrates that this is indeed the case. As can be seen in Fig. 5 (g-h), OA decreases with increasing O:C ratio with an $R^2$ of 0.7. A similar relationship can be found between normalized WSOC chemical age (Fig. 6) in that WSOC decreases with increasing O:C ratio with a $R^2$ of 0.3. However, it is key to note that this correlation is not derived from individual flights and in fact is not robust in each flight and is rather due to fire-to-fire variation. Figure S5 shows the correlation between normalized OA and chemical age within each fire source, which shows that different fires emit different OA, and OA does not always decrease with chemical age within a single fire

(Kiwah fire and Rabbitfoot fire). Therefore, we believe that different OA is caused by fire properties or fast chemistry near the source, but that these markers (O:C, toluene:benzene) provide a significant correlation with the amount of organic aerosol observed in various plumes.

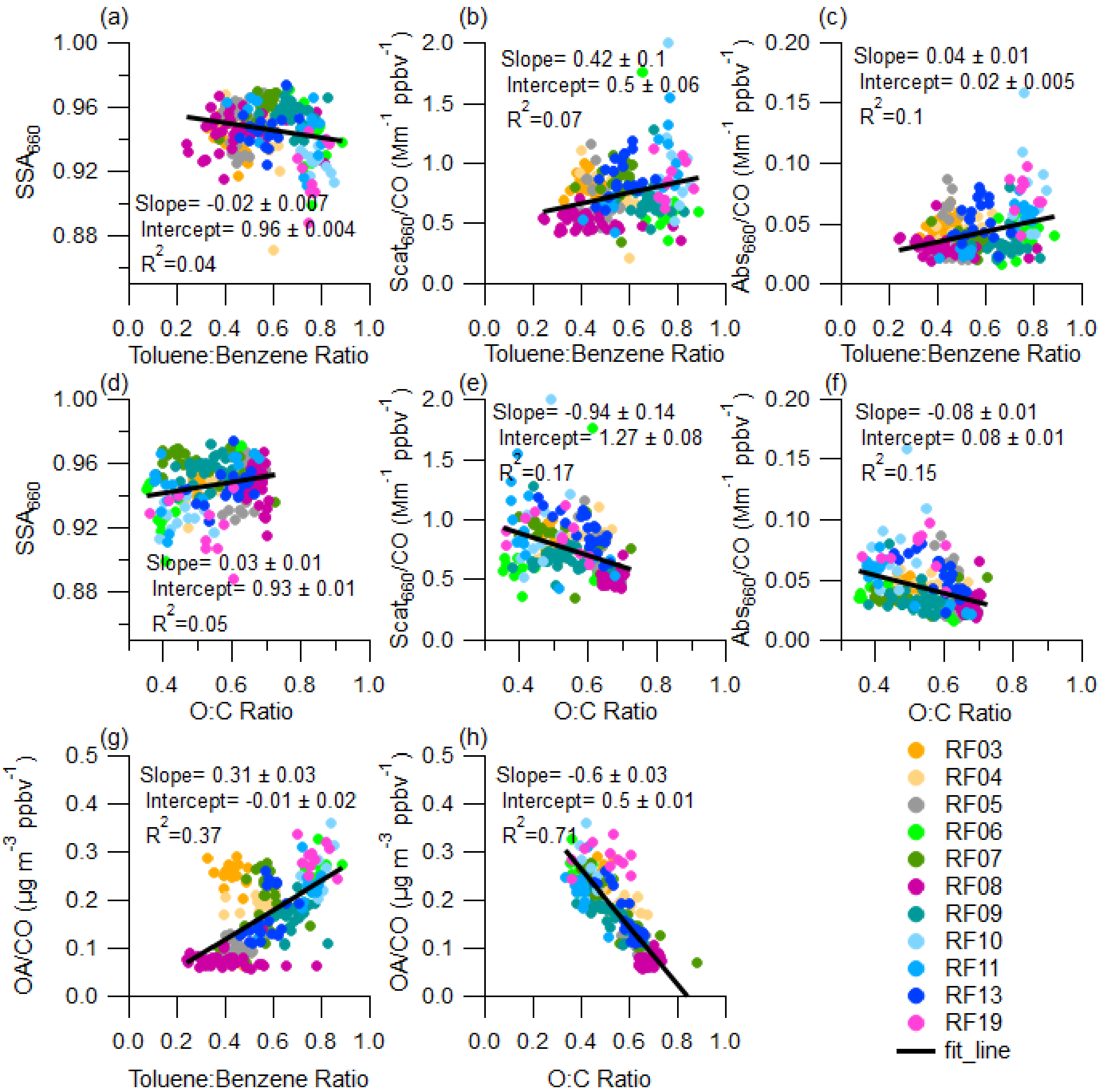

**Figure 5: Plume integrated optical properties at 660 nm and normalized OA variation with chemical age. Top panels show (a) SSA, (b) Scattering, and (c) absorption variation with toluene:benzene ratio. Middle panels show (d) SSA, (e) Scattering, and (f) absorption variation with O:C ratio. Bottom panels show plume integrated normalized OA variation with (g) toluene:benzene ratio and (h) O:C ratio.**

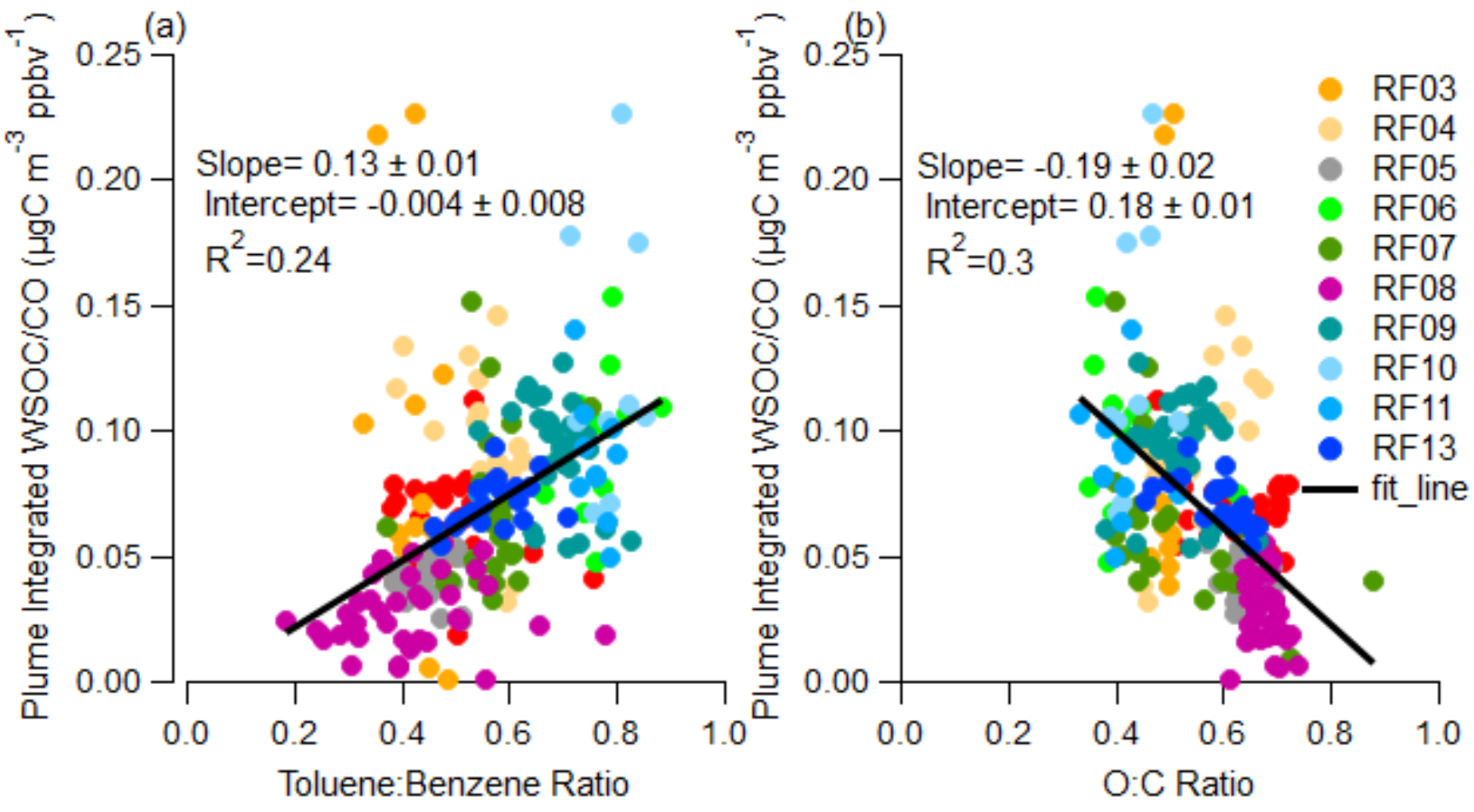

**Figure 6: Plume integrated normalized WSOC variation with (a) toluene:benzene ratio and (b) O:C ratio.**

The behavior of plume-integrated normalized OA with altitude and temperature is shown in Fig. 7 a-b. The trend is not significant, but the main reason is that for most flights we transected the plume, and caused the straight lines within the same colored marker. However, it is still clear that the smallest OA was captured in the plumes (RF08) that have highest temperature (~305 K), and larger OA tends to be observed in the colder plumes (RF19). More studies are needed to determine whether OA is evaporated in high temperature plume, but it's beyond the scope of this work.

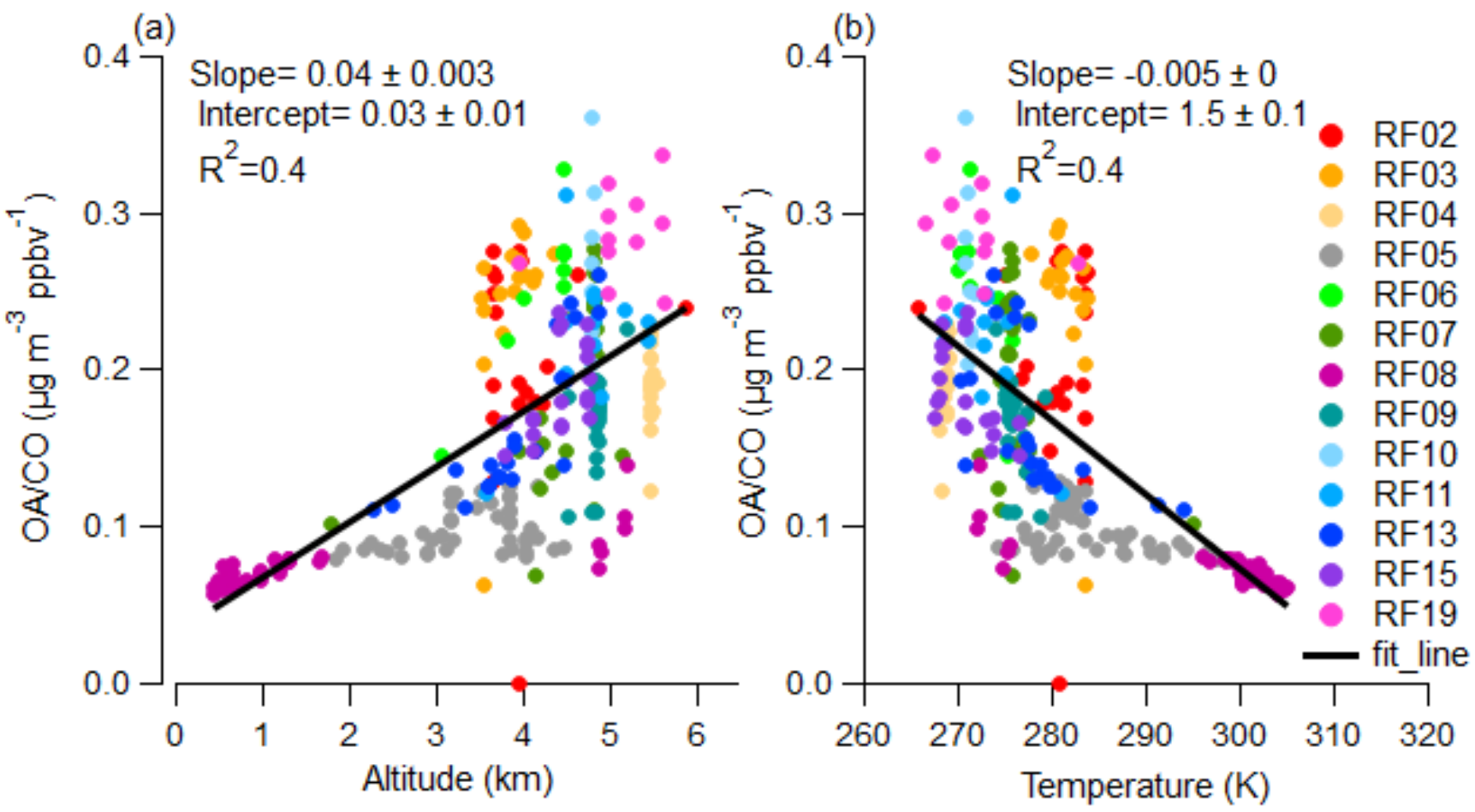

**Figure 7: Plume integrated normalized OA variation with (a) altitude and (b) temperature.**

RF05 and RF08 were chosen as case studies, to observe the optical properties of highly aged aerosol from multiple fire sources to see if the optical properties of this aerosol were similar to those observed in the plume sampling of individual fires at large chemical or physical age. RF08 was a flight through the Central Valley of California where aged smoke from multiple fires that had settled into the valley was measured while RF05 was a flight in which smoke

from several California fires was observed in California, Oregon and Idaho roughly 300~600 miles from the fires (flightpaths are shown in Fig. 1). $MAC_{BC660}$, CO mixing ratio, MCE, toluene:benzene ratio, O:C ratio and $SSA_{660}$ are displayed in Fig. 8a and 8b. The mixing ratio of CO is relatively low in these aged dilute smoke plumes vs. the plumes near the sources analyzed earlier. CO mixing ratio is used as an indicator of smoke when it exceeds 150 ppb. 1-minute averages of $MAC_{BC660}$ are calculated to reduce noise. Therefore, the 1-minute-averages for MCE, toluene:benzene ratio and O:C ratio were also calculated, and all the negative values were removed. As shown in Fig. 8, the smallest toluene:benzene ratio is ~0.35 in RF05, and is ~0.16 in RF08, while the largest O:C ratio is ~0.7 in both RF05 and RF08, which indicates these two cases indeed captured plumes that appear chemically aged compared with the smallest toluene:benzene ratio (0.33) and the largest O:C ratio (0.88) in near-fire measurements shown in Fig. 4.

In RF05 (Fig. 8a), the weighted average O:C ratio over the entire flight was 0.64, the average MCE was 0.82 with a standard deviation of 0.1, the toluene:benzene ratio averaged 0.45 with a standard deviation of 0.05, and the $SSA_{660}$ averaged 0.95 with a standard deviation of 0.01. MCE has a few points because all the negative values were removed, where either CO or CO2 is smaller than the background (CO < 150 ppb). $MAC_{BC660}$ varied from 8.9 $m^2$ $g^{-1}$ to 15.7 $m^2$ $g^{-1}$ with an average of 11.7 $m^2$ $g^{-1}$ and a standard deviation of 1.38 $m^2$ $g^{-1}$. The reasonably large variation of MCE may be caused by variability in the burn conditions of different fire sources, but the overall conclusion is that these emissions, which were measured 300 to 600 miles away, have a very similar $MAC_{BC660}$ to that of the near-source flights.

The RF08 (Fig. 8b) results are similar to RF05, even though these emissions were smoke of mixed aged from multiple fire sources in the Central Valley. The weighted average O:C ratio was 0.67 over the entire measurement, average MCE was 0.84 with a standard deviation of 0.05, average toluene:benzene ratio was 0.41 with a standard deviation of 0.15, and average $SSA_{660}$ was 0.94 with a standard deviation of 0.01. $MAC_{BC660}$ averaged 10.9 $m^2$ $g^{-1}$ with a standard deviation is 2.24 $m^2$ $g^{-1}$. There are several extreme values that exist in the dataset, probably because of time-alignment issues caused by variation in the dilution rate of the SP2 which cannot be totally eliminated from the 1-minute average. In addition, the smoke from RF08 (Fig. 8b) is split into four regions based on observed CO mixing ratios, and integrated $MAC_{BC660}$ is calculated for each region (purple star marker). Region edges are represented by blue dashed lines. Region integrated $MAC_{BC660}$ is relatively stable with an average value of 10.2 $m^2$ $g^{-1}$ and a standard deviation of 0.6 $m^2$ $g^{-1}$.

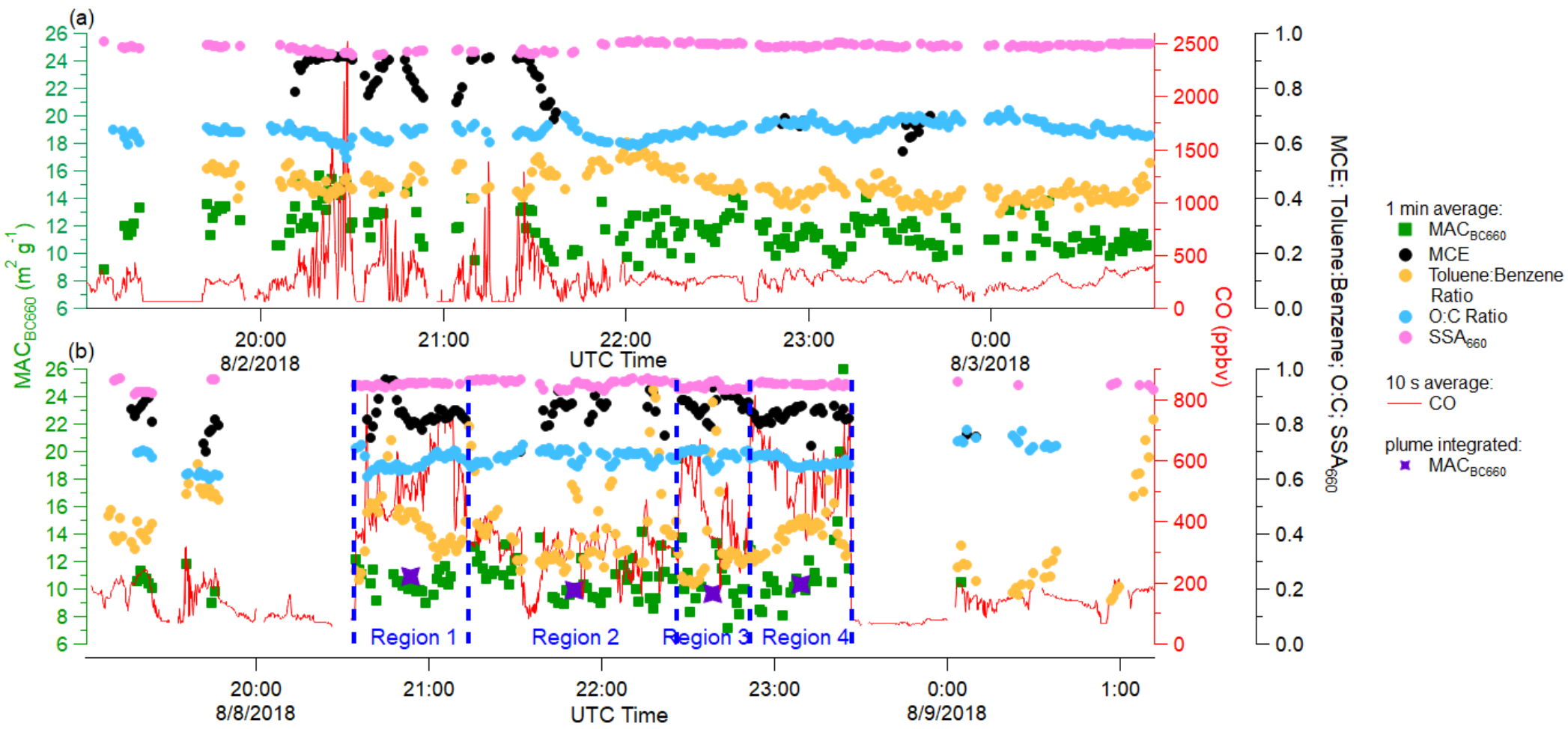

**Figure 8: Time series of plume properties during (a) RF05, and (b) RF08(Central Valley of California). Different square and round markers indicate 1 min averages of different variables as shown in the legend, and the red solid line represents 10 s averages of the mixing ratio of CO. Purple stars in RF08 indicate region integrated MAC$_{BC660}$ (individual regions are separated based on the concentration of CO, and indicated by blue dashed lines).**

The behavior of plume-integrated MAC$_{BC660}$ with altitude and temperature is shown in Fig. 9 a-b. MAC$_{BC660}$ shows little correlation with altitude or temperature, even though there is a large range of both (altitude 500 m to 6 km, temperature 270 K to 305 K). To assess the impact from dilution, the relation between MAC$_{BC660}$ and $\Delta$CO is shown in Fig. 9c. The MAC$_{BC660}$ may decrease slightly with dilution (lower $\Delta$CO), but the correlation is very poor. Neither altitude, temperature, or dilution appear to have a dramatic impact on the MAC$_{BC660}$.

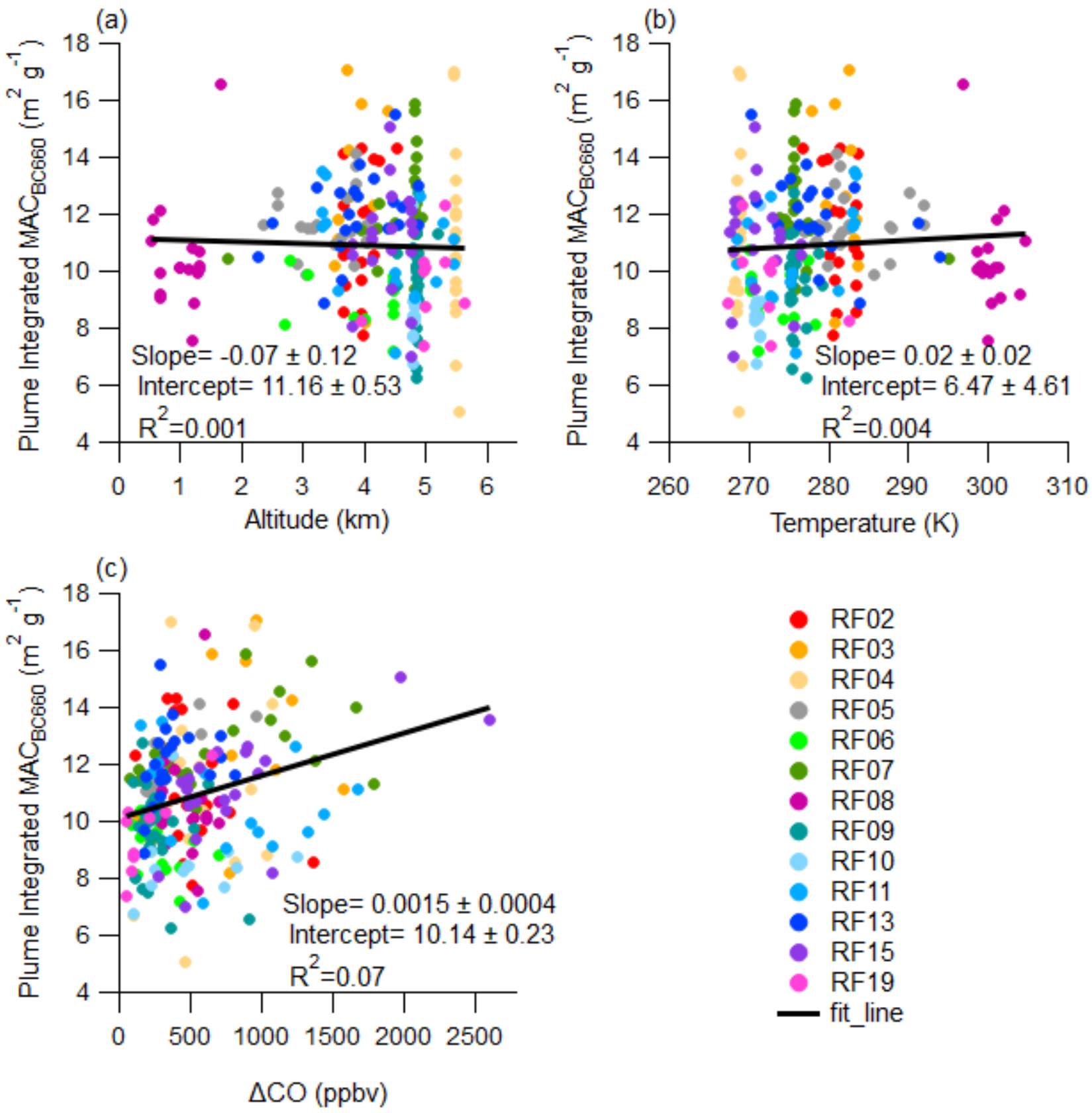

**Figure 9: Plume integrated $MAC_{BC660}$ variations with (a) Altitude, (b) Temperature, and (c) ΔCO.**

### 3.1.4 Contribution of Brown Carbon Versus Lensing at 660 nm

Many previous studies of BrC assume that BrC does not absorb significant amounts of light at long wavelengths (532~705 nm) (Wonaschütz et al., 2009; Lack et al., 2012a; Taylor et al., 2020; Zeng et al., 2021, Zhang et al., 2022). In this study, a PILS system was used to quantify the absorption of light for water-soluble BrC at 660 nm. This absorption is not likely caused by traditional BC, which is insoluble and will be removed by the 0.2 µm filter in the PILS (Peltier et al., 2007; Zeng et al., 2021).

One aim was to investigate if absorption enhancement at 660 nm was primarily due to the lensing effect or due to absorption from BrC. The fraction of non-BC absorption from BrC at 660 nm was calculated by Eq. 5. To convert the measured light absorption by water-soluble organics into total BrC absorption in the ambient, it had to be multiplied by two factors. The first factor converts absorption from water-soluble BrC into absorption from total BrC. This factor is obtained by taking the ratio between total particulate organic mass and water-soluble particulate organic mass (OM:WSOC). Water-soluble organic mass is calculated from the PILS WSOC data using a WSOM:WSOC (water-soluble organic mass : water-soluble organic carbon) ratio of 1.6 (Duarte et al., 2015 & 2019). Ambient organic

mass is measured by the AMS or calculated from the particle size distributions measured by the UHSAS assuming the particle mass all comes from organic material with a particle density of 1.4 g cm$^{-3}$. Both methods are used and compared in this paper. The second factor accounts for the fact that particles absorb more light than the same substance in the bulk liquid phase. Here we use Mie theory to convert absorption from BrC in aqueous solution to the absorption from BrC particles in the ambient (Liu et al., 2013; Zeng et al., 2020). The complex refractive index (m = n+ $i$k) was put into a Mie code to obtain the absorption efficiency (Q), and further used to calculate the absorption coefficient by Eq. 9 (Liu et al., 2013). The real part of the refractive index (n) is set to be 1.55, and the imaginary part is calculated by using Eq. 10 (Liu et al., 2013),

$$\beta\left(\lambda, D_p\right) = \frac{3}{2} \cdot \frac{Q \cdot WSOC}{D_p \cdot \rho} \qquad (Eq. 9)$$

$$k = \frac{\rho\lambda \cdot H_2O_\beta(\lambda)}{4\pi \cdot WSOC} \qquad (Eq. 10)$$

where $\lambda$ is the wavelength, $D_p$ is the diameter of the particle, $\beta$ is absorption coefficient, $Q$ is absorption efficiency, particle density ($\rho$) is set to be 1.4 g cm$^{-3}$, $WSOC$ is the mass concentration of WSOC (μgC m$^{-3}$) measured by the PILS, and $H_2O_\beta(\lambda)$ is the water-soluble light absorption coefficient measured by PILS. The plume averaged particle size distribution was used in the calculation, then the absorption coefficient was calculated for each size bin of UHSAS to obtain the most accurate Mie factor for each plume.

The average OM:WSOC factor based on the UHSAS (UHSAS factor) for all the plumes is 2.36 with a standard deviation is 1.17. The averaged OM:WSOC based on the AMS (AMS factor) is 1.63 with a standard deviation of 0.74. The average Mie factor at 660 nm is 1.47 (standard deviation of 0.13), which is close to the factor of 1.36 found by Zeng et al. (2022) based on FIREX data. The Mie factor at 405 nm based on the WE-CAN data is also calculated, with an average of 1.83, which is similar to the factor that Zeng et al. (2022) determined at 405 nm (1.7) based on FIREX and Liu et al. (2013) determined at 450 nm (1.9) based on measurements in Atlanta.

Sensitivity tests were done on these factors by choosing reasonable ranges of particle density (1.1 g cm$^{-3}$, 1.4 g cm$^{-3}$ and 1.7 g cm$^{-3}$) and WSOM:WSOC ratio (1.5, 1.6 and 1.8) (Duarte et al., 2015 & 2019; Finessi, et al., 2012; Sun et al., 2011) (Table 1). Particle density only affects the Mie factor and UHSAS factor, while WSOM:WSOC ratio affects the AMS factor and UHSAS factor. As shown in Table 1, the impact of particle density on the Mie factor (both at 660 nm and 405 nm) is negligible, WSOM:WSOC is the only component that affects the AMS factor (ranging from 1.48 to 1.73), while the UHSAS factor is much more sensitive (ranging from 1.65 to 3.06) to both particle density and WSOM:WSOC. Overall, Table 1 demonstrates that none of the factors other than the UHSAS factor are sensitive to the exact parameters chosen for the calculation, giving confidence that the results presented are robust.

**Table 1: Average values and standard deviation of AMS factor, Mie factor at 660 nm and UHSAS factor for all the integrated plumes when using different particle density and WSOM:WSOC ratio. Unit of particle density is g cm$^{-3}$.**

| Factor | Particle Density | WSOM:WSOC | Average | Standard Deviation |
|---|---|---|---|---|
| AMS factor | N/A | 1.5 | 1.73 | 0.79 |
| | N/A | 1.6 | 1.63 | 0.74 |
| | N/A | 1.8 | 1.48 | 0.79 |

| | | | | |
|---|---|---|---|---|
| Mie factor at 660 nm | 1.1 | N/A | 1.47 | 0.13 |
| | 1.4 | N/A | 1.47 | 0.13 |
| | 1.7 | N/A | 1.47 | 0.13 |
| Mie factor at 405 nm | 1.1 | N/A | 1.83 | 0.89 |
| | 1.4 | N/A | 1.83 | 0.89 |
| | 1.7 | N/A | 1.83 | 0.89 |
| UHSAS factor | 1.1 | 1.8 | 1.65 | 0.82 |
| | 1.4 | 1.6 | 2.36 | 1.17 |
| | 1.7 | 1.5 | 3.06 | 1.52 |

Figure 10 shows the time evolution of the fraction of non-BC absorption from BrC at 660 nm for the biomass burning plumes encountered during WE-CAN. Assuming a MAC of the BC core of 6.3 $m^2$ $g^{-1}$, BrC contributes roughly the same amount of absorption at 660 nm as lensing (62% UHSAS method, 46% AMS method). This means that 19% (based on the AMS) to 26% (based on the UHSAS) of the total absorption at 660 nm comes from BrC. When different particle density and WSOM:WSOC ratios are considered (top and bottom whiskers, as well as red and blue dashed lines), the fraction of non-BC absorption is 43-80% for the UHSAS approach and 41-49% for the AMS approach based on different OM:OC and density. While there is considerable variability between flights, a rule of thumb that roughly half of the non-BC absorption at red wavelengths is from absorbing organic material seems reasonable. To our knowledge, this is the first attempt to differentiate between lensing and absorbing organics in the red wavelengths. This approach assumes that water insoluble BrC has the same refractive index as water soluble BrC.

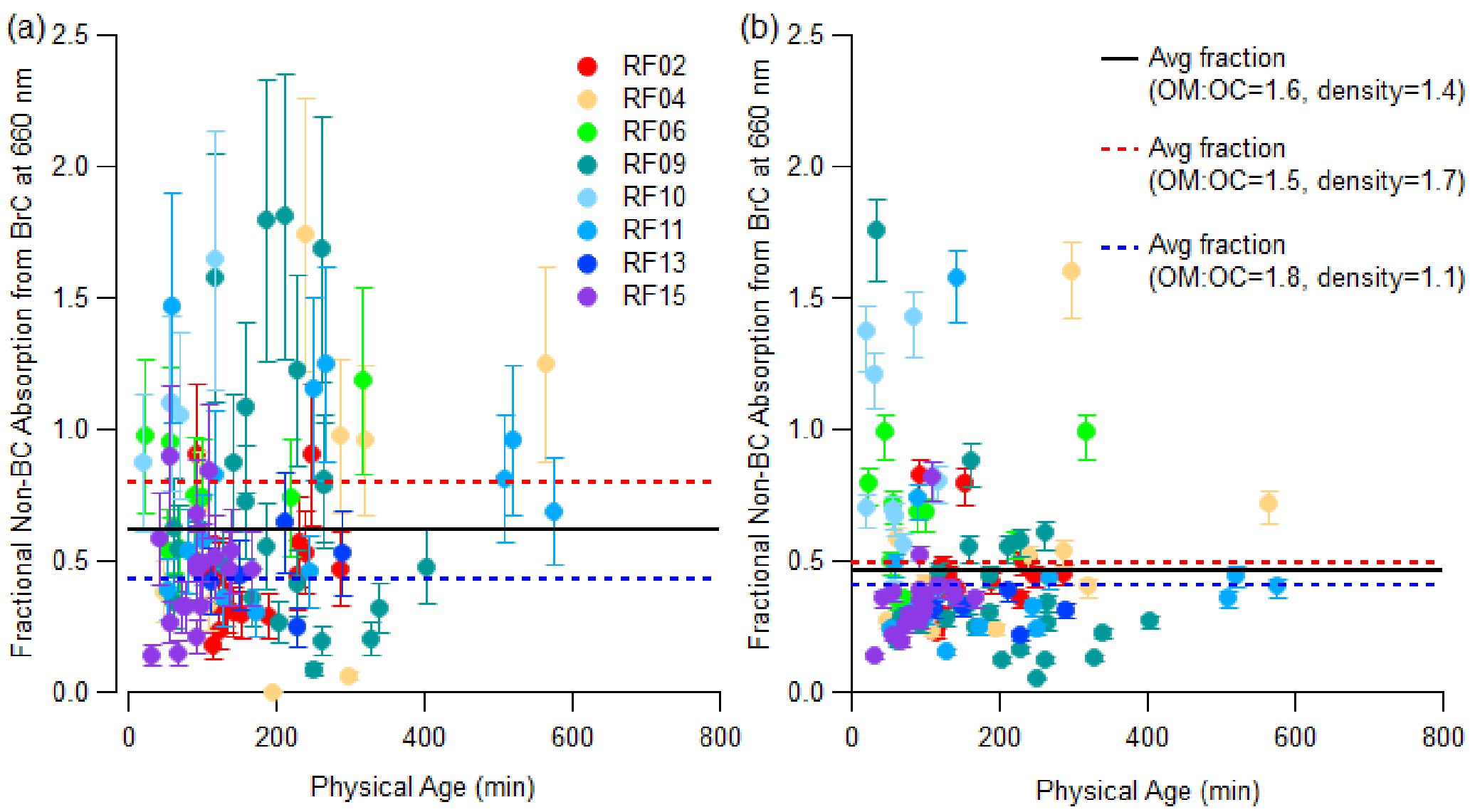

**Figure 10: Time evolution of the fraction of non-BC absorption from BrC at 660 nm (a) with UHSAS and Mie factor, (b) with AMS and Mie factor. The UHSAS was not available in RF03. Markers were calculated using a density of 1.4 g cm$^{-3}$ and WSOM:WSOC ratio of 1.6. The top whiskers represent sensitive test values using a density of 1.7 g cm$^{-3}$ and WSOM:WSOC ratio of 1.5, while the bottom whiskers represent sensitive test values using a density of 1.1 g cm$^{-3}$ and WSOM:WSOC ratio of 1.8. The averaged fraction of non-BC absorption from BrC from all the plumes are shown in black solid lines, while the range of this result from sensitivity tests are shown in red and blue dashed lines.**

### 3.1.5 Aging of Water Soluble BrC at 660 nm

To further analyze the evolution of BrC at 660nm in wildfire emissions, the MAC of water-soluble BrC at 660 nm ($MAC_{ws_BrC660}$) was calculated by taking the ratio of water-soluble light absorption ($\beta_{ws_BrC660}$) and WSOC provided by the PILS (Eq. 4). Similar to $MAC_{BC660}$, $MAC_{ws_BrC660}$ is relatively flat with physical age (Fig. 11a), with an average of 0.06 m$^2$ g$^{-1}$ and a standard deviation of 0.04 m$^2$ g$^{-1}$, but most of the plumes measured were less than 10 hours old. Interestingly, the fit lines for correlations with markers of chemical age suggest that $MAC_{ws_BrC660}$ tends to be larger when these markers indicate a more oxidized plume (Fig. 11 b-c), which is distinct from what has been found in previous studies that BrC at shorter wavelengths decays with chemical age. The trend of increasing $MAC_{ws_BrC660}$ is not clear in each flight and is only observed when properties are compared between fires. While the correlation coefficients are low, it can be stated that $MAC_{ws_BrC660}$ is consistently larger in more oxidized plumes and there is not a decrease with increased oxidation or chemical aging. Given that all the observed smoke plumes were of similar physical ages, this again leads to the idea that these properties are the result of different emissions or fast chemistry that occurs before the plumes are first observed by the aircraft.

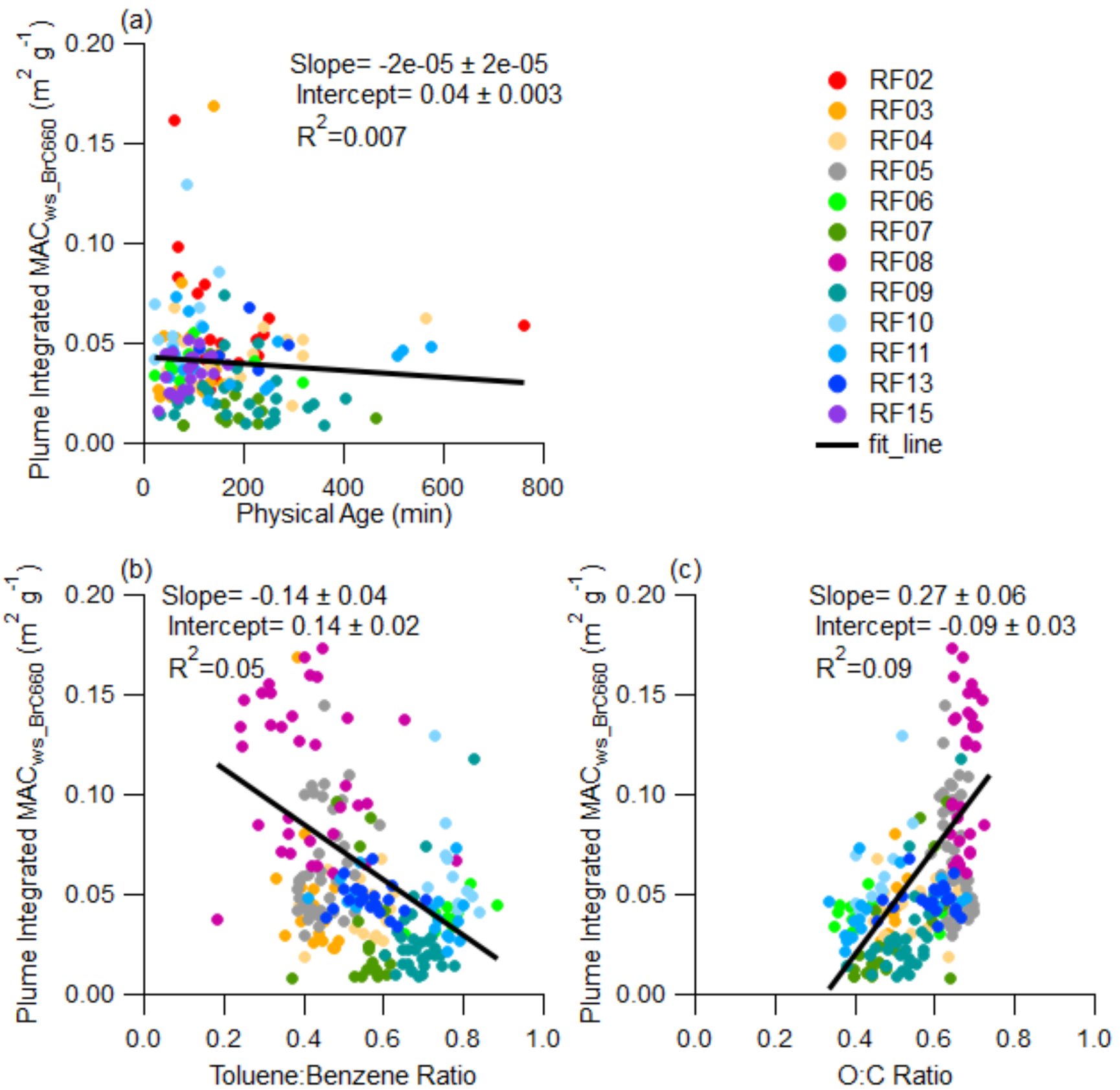

**Figure 11: Plume integrated $MAC_{ws_BrC660}$ variations with (a) Physical Age; (b) toluene:benzene ratio and (c) O:C ratio. RF05 and RF08 were measuring mixed aged smokes, during which the fire source cannot be identified, and therefore the physical age is unavailable for both flights.**

The total MAC of BrC and lensing at 660 (measured by PAS, $MAC_{BrC+lensing_660}$) is also compared with markers of chemical age (Fig. 12). $MAC_{BrC+lensing_660}$ includes lensing, water-soluble BrC, and water-insoluble BrC. $MAC_{BrC+lensing_660}$ also shows an increasing trend with chemical oxidation, which is consistent with the trend in $MAC_{ws_BrC660}$. Again, it is important to point out that while there is a weak trend for some individual flights, much of the trend of increasing MAC at 660 nm results from combining data from all the fires and much of the trend may be caused by emissions from different fires having different properties. Despite this, it is clear that plumes that appear chemically older by either the toluene:benzene ratio or the O:C ratio actually have larger MAC's in the red wavelength than plumes that appear chemically younger, a result that is exactly the opposite of the bleaching of BrC often seen at shorter wavelengths. In addition, these chemically "older" plumes have less BC, OA, and WSOC along with smaller bulk absorption. That is to say, when chemical markers indicate an "old" plume, the fire tends to emit darker (Fig. 11 and Fig. 12) organic aerosol, but less of it (Fig. 5 g-h and Fig. 6) resulting in less bulk absorption (Fig. 5c and 5f). The mean value of $MAC_{BrC+lensing_660}$ is 0.11 $m^2$ $g^{-1}$ (with a standard deviation of 0.06), which is larger than the 0.06 average

of $MAC_{ws_BrC660}$, a result we have attributed to the lensing effect, but which could also partially be the result of water-insoluble BrC having a higher MAC than water-soluble BrC.

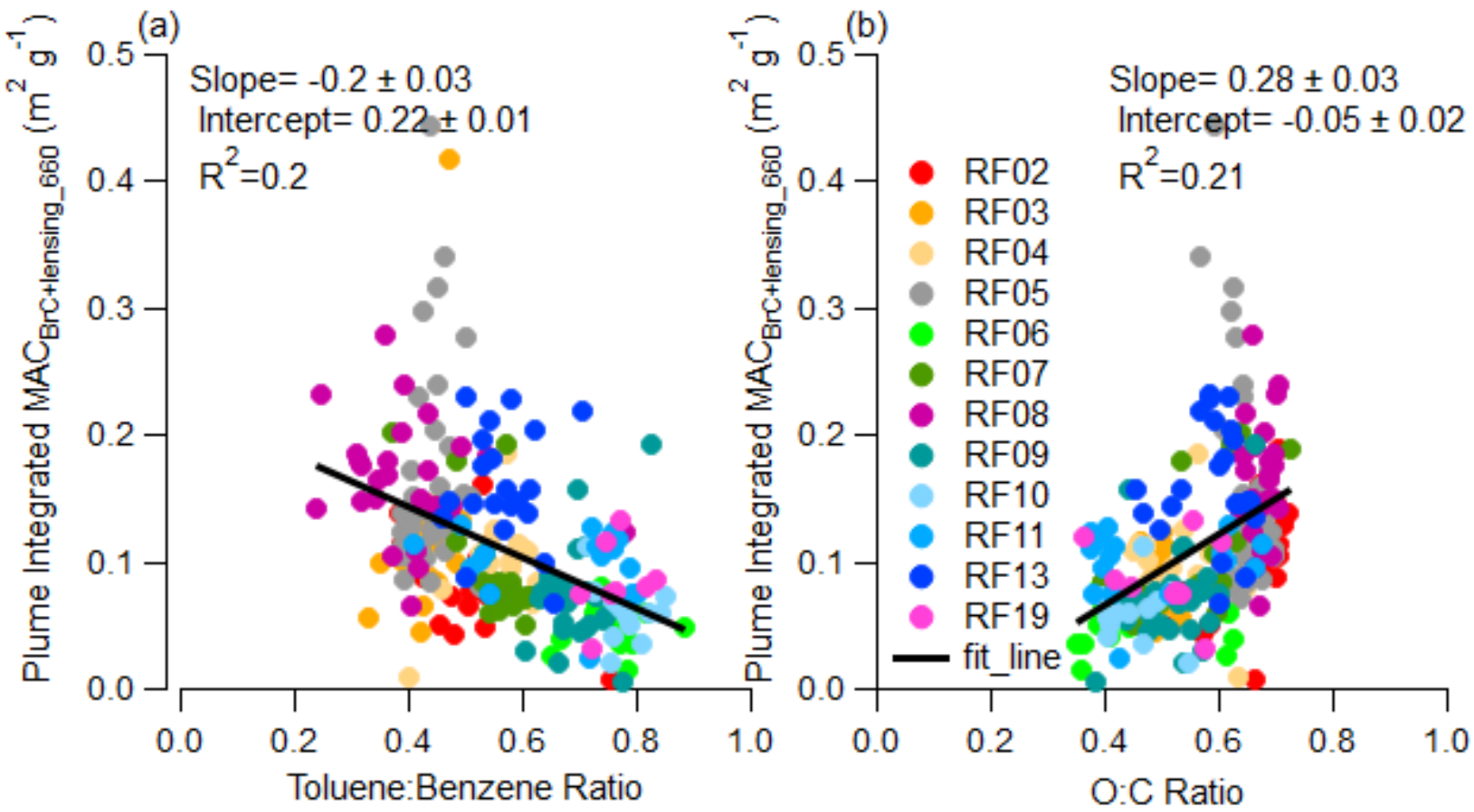

**Figure 12: Plume integrated $MAC_{BrC+lensing_660}$ variations with (a) toluene:benzene ratio and (b) O:C ratio.**

The ratio of BC:OA has been shown to correlate to the optical properties of biomass burning aerosol (Pokhrel et al, 2016; Saleh et al., 2014). During WE-CAN, $MAC_{BrC+lensing_660}$ showed an increasing trend with increasing BC:OA ratio (Fig. S6), which is similar to Saleh et al. (2014), who found that the imaginary part of the refractive index of OA at 550 nm increases with the BC:OA ratio. This consistency suggests that MAC of BrC at 550 nm and 660 nm have the same behavior in that BrC grows more absorbing as the BC:OA ratio increases.

### 3.2 Characteristics of BrC at 405 nm

#### 3.2.1 Behavior of $MAC_{BrC+lensing_405}$

Palm et al. (2020) utilized data from both WE-CAN and Monoterpene and Oxygenated aromatic Oxidation at Night and under LIGHTs (MOONLIGHT) campaigns and found that evaporated biomass-burning POA is the dominant source of the biomass-burning SOA in wildfire plumes, which happened in the first a few hours after emission. They also found that for those SOA that formed from oxidation, phenolic compounds contribute 29 ± 15% of BrC absorption at 405 nm. In this section, characteristics of BrC were also analyzed at 405 nm to understand if the behavior was similar or different to BrC decay at 660 nm. BrC at 405 nm was calculated in the same way that it was calculated at 660 nm, following Eq. 3, and therefore it also has a contribution from the lensing effect. Figure 13 shows the plume integrated $MAC_{BrC+lensing_405}$ variations with (a) Physical Age and (b) MCE. Similar to 660 nm, $MAC_{BrC+lensing_405}$ varies from fire to fire and no clear trend can be found with increasing physical age or MCE. Similar behavior was also observed in Western wildfires at 405 nm in FIREX-AQ (Zeng et al., 2022). The $MAC_{BrC+lensing_405}$, varies from 0.08 $m^2$ $g^{-1}$ to 1.6 $m^2$ $g^{-1}$ with a mean value of 0.59 $m^2$ $g^{-1}$ and a standard deviation of 0.19. The largest values are from RF05, the flight through California, Oregon, and Idaho, where aged smoke from different fires was mixed. The large $MAC_{BrC+lensing_405}$ in RF05 is related to relatively small OA (Fig. 5 g-h), which occurred when the

plane left the smoke-filled boundary layer during RF05. If we exclude $MAC_{BrC+lensing_405}$ from RF05, the values range from 0.08 $m^2$ $g^{-1}$ to 1.09 $m^2$ $g^{-1}$, but still have a mean value of 0.59 $m^2$ $g^{-1}$ and a standard deviation of 0.15. Again, we note that this value includes the contribution of lensing. Despite this, our results lie in the same range as those measured without the contribution of lensing of 0.31 ± 0.09 $m^2$ $g^{-1}$ measured in CLARIFY-2017 (Taylor, 2020), 0.13-2.0 $m^2$ $g^{-1}$ measured in FIREX-AQ (Zeng et al., 2022), and 0.25-1.18 $m^2$ $g^{-1}$ measured in ORACLES (Zhang et al., 2022).

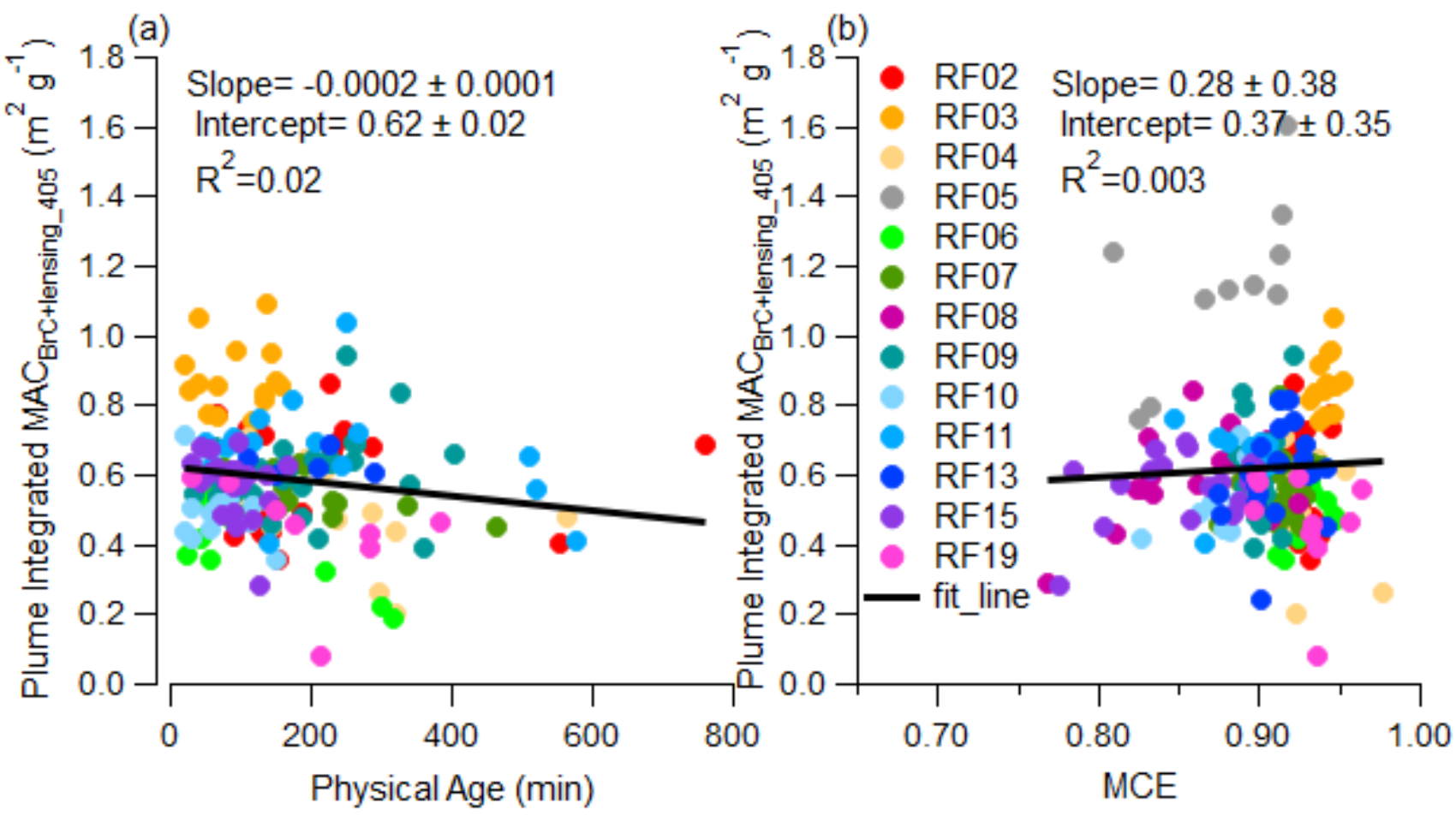

**Figure 13: Plume integrated $MAC_{BrC+lensing_405}$ variations with (a) Physical Age and (b) MCE**

Figure 14 shows the behavior of brown carbon at 405 nm with markers of chemical aging. Very weak or non-trends are observed. If there is any trend, it is a slight increase in $MAC_{BrC+lensing_405}$ with decreasing toluene:benzene ratio, which is consistent with the results for $MAC_{BrC+lensing_660}$ (Fig. 12) and $MAC_{ws_BrC660}$ (Fig. 11). The flat or slightly increasing trend with increasing oxidation shows that the decrease in total aerosol absorption with markers of chemical age is due to a decrease in OA (Fig. 5 g-h) because the BrC is actually darker in oxidized plumes at all wavelengths. It is important to remember that most of the trends observed in WE-CAN are caused by emissions from different fires versus variations within a fire, which tend to be quite small. Consistent results for the behavior of $MAC_{BrC}$ at different wavelengths derived using different instruments (PAS and PILS) is further evidence that BrC decay does not occur in the WE-CAN dataset, or at least that plume integrated results cannot capture the BrC decay that might be occurring at the edges of the plume. While further research focused on the edge of the plumes, which often appear highly oxidized, is needed, this is beyond the scope of the current work. Despite this, it is the plume integrated results that are relevant for climate impacts and for comparison to model output, discussed in the following section.

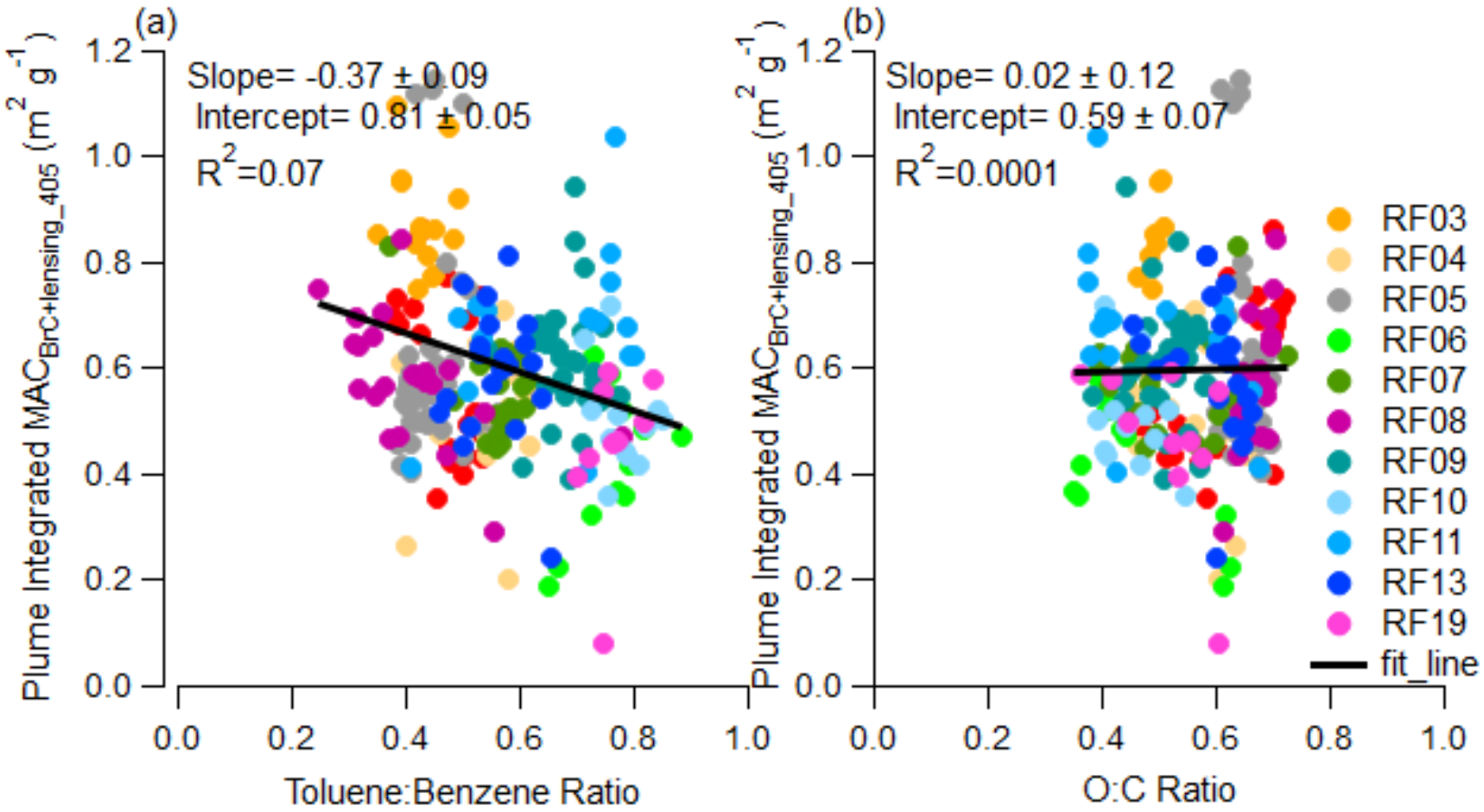

**Figure 14: Plume integrated MAC$_{BrC+lensing_405}$ variations with (a) toluene:benzene ratio and (b) O:C ratio**

RF05 and RF08 are presented as case studies again to investigate the behavior of MAC$_{BrC+lensing_405}$ in mixed plumes emitted from different fire sources. Figure 15 is similar to Fig. 8, but with MAC$_{BrC+lensing_405}$ instead of MAC$_{BC660}$. For the case of RF05 (Fig. 15a) MAC$_{BrC+lensing_405}$ varied from 0.36 $m^2$ $g^{-1}$ to 1.52 $m^2$ $g^{-1}$ with an average of 0.66 $m^2$ $g^{-1}$ and a standard deviation of 0.26 $m^2$ $g^{-1}$, the SSA$_{450}$ averaged 0.94 with a standard deviation of 0.02, which is similar to SSA$_{660}$. The MAC$_{BrC+lensing_405}$ is larger when CO mixing ratio is higher, but does not have a significant correlation with any other variables shown in Fig. 15. For the case of RF08 (Fig. 15b) MAC$_{BrC+lensing_405}$ is more stable than in RF05, and varied from 0.25 $m^2$ $g^{-1}$ to 0.88 $m^2$ $g^{-1}$ with an average of 0.59 $m^2$ $g^{-1}$ and a standard deviation of 0.14 $m^2$ $g^{-1}$, the SSA$_{450}$ average was 0.95 with a standard deviation of 0.01. The regional integrated MAC$_{BrC+lensing_405}$ is even more stable with an average value of 0.59 $m^2$ $g^{-1}$ and a standard deviation of 0.07 $m^2$ $g^{-1}$.

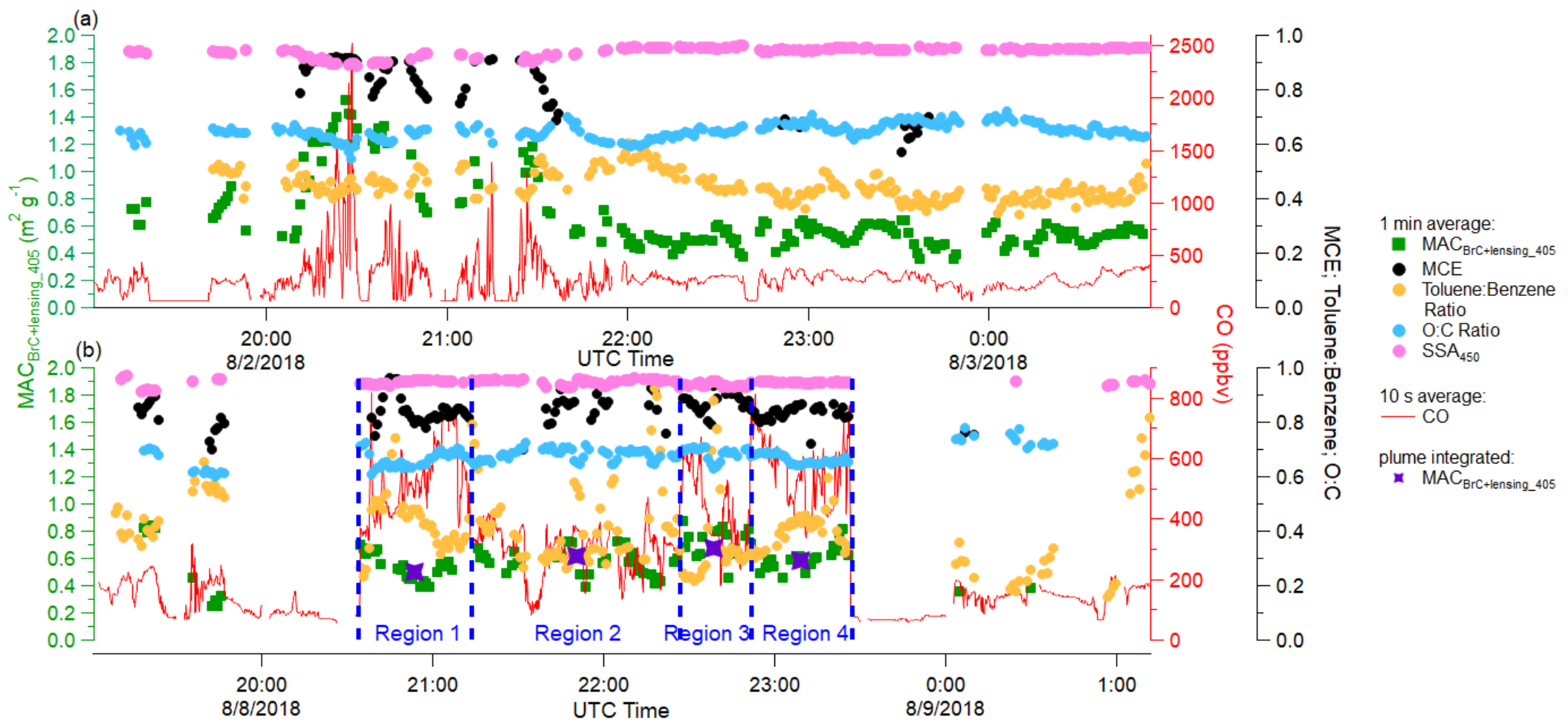

**Figure 15: Time series of plume properties during (a) RF05, and (b) RF08(Central Valley of California). Different square and round markers indicate 1 min averages of different variables as shown in the legend, and the red solid line represents 10 s averages of the mixing ratio of CO. Purple stars in RF08 indicate region integrated $MAC_{BrC+lensing_405}$ (individual regions are separated based on the concentration of CO, and indicated by blue dashed lines).**

When comparing $MAC_{BrC+lensing_405}$ with altitude and temperature (Fig. 16), it has the same behavior with $MAC_{BC660}$ in that $MAC_{BrC+lensing_405}$ stays relatively constant with both altitude and temperature. In addition, when $MAC_{BrC+lensing_405}$ is plotted vs. ΔCO (not shown), no clear change in $MAC_{BrC+lensing_405}$ is seen due to dilution. Therefore, both $MAC_{BrC+lensing_405}$ and $MAC_{BC660}$ do not appear to be affected by altitude or temperature during WE-CAN.

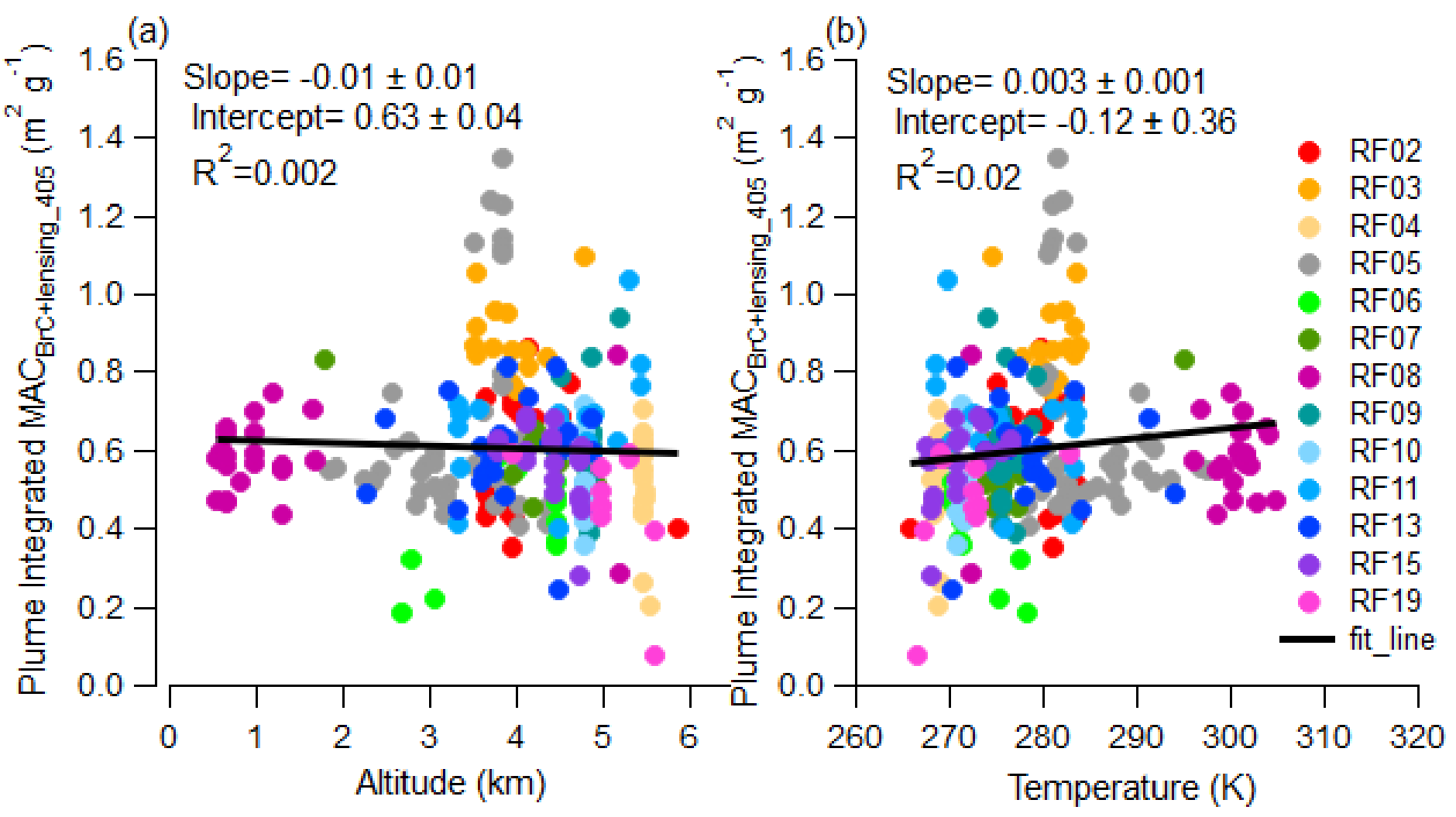

**Figure 16: Plume integrated $MAC_{BrC+lensing_405}$ variations with (a) Altitude and (b) Temperature**

When comparing the relationship between $MAC_{BrC+lensing_405}$ with the BC:OA ratio (Fig. S7), there is not a clear increasing trend, and the correlation is worse than that at 660 nm (Fig. S6). However, the increasing trend still exists in most individual flights (not in RF04 and RF10). The increasing trend is not as clear as in Saleh et al. (2014), most probably because the range of BC:OA ratios observed during WE-CAN (0.007~0.061) is much smaller than that (0.005~0.7) observed in Saleh's work. Even in their work, the increasing trend is not very clear if one only focuses on the region where the BC:OA ratio is less than 0.03. Also, the Saleh et al. results were obtained from laboratory burns and not wildfires, which might also cause a discrepancy.

### 3.2.2 Comparison of WE-CAN Results to Modeling Studies

The BC:OA ratios measured during WE-CAN were utilized in the Saleh et al. parameterization (2014), which provides an imaginary part for the refractive index of BrC ($k_{BrC,\lambda}$) as a function of the BC:OA ratio. The mean BC:OA ratio for each plume was used in the parameterization, which gave an average $k_{BrC}$ of 0.025, 0.013, 0.009, respectively, at 405 nm, 550 nm and 660nm. Mie theory was then used to calculate the MAC for BrC, in which we assumed a real part of the refractive index of 1.7 for BrC (same as Saleh et al., 2014), a volume mean diameter measured for each plume, and a density of 1.4 g $cm^{-3}$. Figure 17 compares the observed $MAC_{BrC+lensing}$ and $MAC_{ws_BrC}$ with the value calculated from the Saleh parameterization with inputs from WE-CAN. In both the observations and the parameterization, the $MAC_{BrC}$ decreases as wavelength increases. However, the Saleh parameterization is always significantly larger than the observations. The $MAC_{BrC}$ from the Saleh parameterization, which does not include lensing effects, is a factor of 3.4 and 2.8 larger than the observed $MAC_{BrC+lensing}$ at 405 nm and 660 nm, respectively. The range of BC:OA ratios during WE-CAN (0.007~0.061) is much smaller than that (0.005~0.7) used in Saleh's work, and the parameterization failed to capture absorbing aerosol properties for this study. The discrepancy could be partly because the data Saleh et al. used for their parameterization comes from controlled laboratory burns and not wildfires, or because of the sensitivity of $MAC_{BrC}$ to density when using the Saleh parameterization. When we increase particle density from 1.4 g $cm^{-3}$ to 1.7 g $cm^{-3}$, the Saleh parameterization median $MAC_{BrC}$ decreases to 1.6 $m^2$ $g^{-1}$ and 0.24 $m^2$ $g^{-1}$, respectively, at 405 nm and 660 nm (a factor of 2.8 and 2.3, respectively, compared to observed $MAC_{BrC}$ at 405 nm and 660 nm). This suggests that the Saleh parameterization overestimates the absorption property of biomass aerosol especially for fresh emitted aerosols and more parameterizations need to be developed. Carter et al. (2021) utilized the Saleh parameterization for BrC absorption in the GEOS-Chem model and also found that the Saleh model overestimated BrC absorption for WE-CAN. It was hypothesized that the overestimation was due to the lack of a bleaching process for BrC in the model and offset part of the overestimation by bringing in bleaching into the model. However, our results show that the overestimation in the model is caused by an incorrect refractive index.

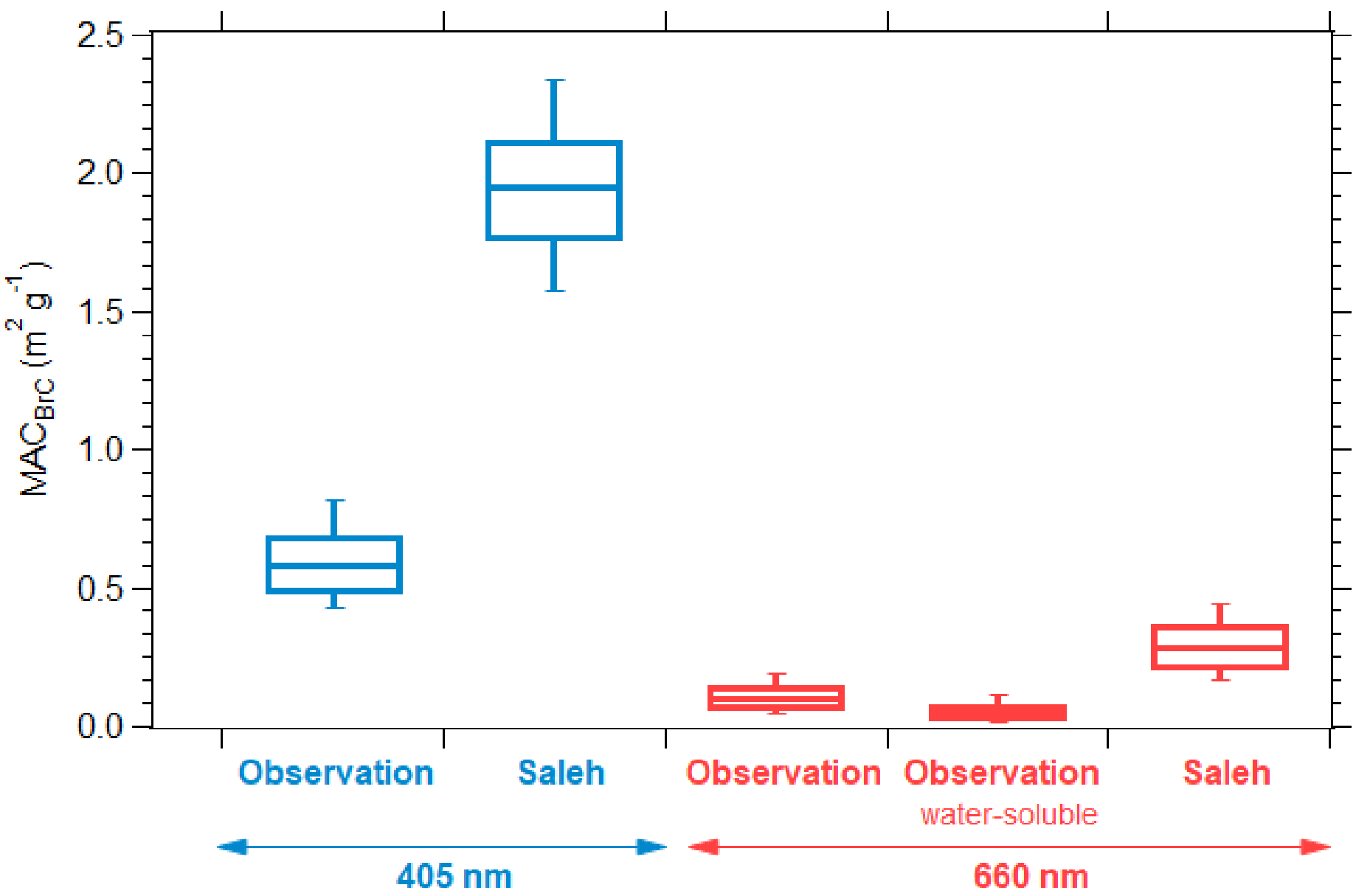

**Figure 17: Boxplot summary for observed and parameterized (Saleh) $MAC_{BrC}$ at 405 nm (blue) and 660 nm (red). On each box the central line represents the median, the top and bottom edge represents 75% and 25%, the top and bottom whisker represents 90% and 10%.**

## 4 Conclusion

In this study, we presented results that help us better understand the ability of wildfire aerosol emissions to absorb visible light and how those properties change after emission. We presented mass absorption coefficients (MAC) for black and brown carbon from Western United States wildfires measured during the WE-CAN campaign at both short and long visible wavelengths ($MAC_{BC660}$, $MAC_{BrC+lensing_660}$, $MAC_{ws_BrC660}$, $MAC_{BrC+lensing_405}$). We also investigated single scattering albedo (SSA), total scattering and total absorption at 660. We observed that the mass absorption coefficient of black carbon stayed relatively constant across all plumes measured and at all physical ages (ages up to 15 hours observed), with an averaged $MAC_{BC660}$ of 10.9 ± 2.1 $m^2$ $g^{-1}$ (average ± standard deviation). This average showed no variation with altitude or temperature, and we saw no evidence that $MAC_{BC660}$ is correlated to MCE. Even in highly aged plumes with emissions mixed from multiple fires (RF05 and RF08), the $MAC_{BC660}$ is similar in magnitude and consistency with an average of 11.3 ± 1.8 $m^2$ $g^{-1}$. Both the fact that this MAC is significantly larger than the MAC for uncoated BC (often cited to be ~ 6.3 $m^2$ $g^{-1}$) and the fact that the MAC remains relatively constant across different fires and different plume ages are key insights that can improve models of aerosol optical properties in wildfire emissions.

We find that total organic aerosol (OA) and water-soluble organic carbon (WSOC) are strongly correlated with markers of chemical age. OA and WSOC (both normalized to CO) decrease with decreasing toluene:benzene ratio or increasing O:C ratio. However, this phenomenon is observed in variations between different fire sources rather

than during the aging of individual fire plumes. We interpret this variability to mean the fires either had different emission ratios of toluene:benzene and O:C or the smoke underwent rapid secondary chemistry prior to the first plume pass in WE-CAN. Regardless, the correlations are relatively strong ($R^2$ of 0.24 to 0.71) and provide a potential link between chemical markers and total organic aerosol amounts across a wide range of fires. While OA and WSOC decrease with decreasing toluene:benzene or increasing O:A, $MAC_{BC660}$ actually shows a weak increasing trend with these same markers of aging, showing that while the total amount of organic aerosol is decreasing, the ability of the organic to absorb per mass is staying relatively constant, or even increasing.

Through a novel use of PILS data, we find that BrC contributes 41-80% of non-BC absorption at 660 nm (assuming 6.3 $m^2\ g^{-1}$ as the MAC of BC core at 660 nm). BrC contributes, on average, 26% of total absorption, but the absorption cross section of water-soluble BrC is relatively small at 660 nm, with a $MAC_{ws_BrC660}$ of 0.06 ± 0.04 $m^2\ g^{-1}$, which does not change with physical age. The average $MAC_{BrC+lensing_660}$ derived from the PAS (which includes both brown carbon absorption and lensing of black carbon) is 0.11 ± 0.06 $m^2\ g^{-1}$.

In the blue visible wavelengths where brown carbon is more often thought about, $MAC_{BrC+lensing_405}$ is 0.59 ± 0.19 $m^2\ g^{-1}$ and shows little variation with physical age or MCE. There are weak increasing trends in all the $MAC_{BrC}$ data we obtained ($MAC_{ws_BrC660}$, $MAC_{BrC+lensing_660}$ and $MAC_{BrC+lensing_405}$) with markers of chemical age (toluene:benzene, O:C), while bulk absorption of total aerosol decreases with these same markers of chemical age. In highly aged plumes from multiple fires (RF05 and RF08), the $MAC_{BrC+lensing_405}$ has an average value of 0.63 ± 0.2 $m^2\ g^{-1}$, suggesting that brown carbon remains significantly absorbing even at relatively longer ages.

Utilizing a common parameterization for BrC refractive index from Saleh et al. (2014), with measured inputs for the BC:OA ratio and particle size, we calculated the theoretical $MAC_{BrC660}$ and $MAC_{BrC405}$, and they were 2.3~3.4 times larger than the measured $MAC_{BrC+lensing}$ during WE-CAN. While this discrepancy has been resolved previously by implementing bleaching into model schemes, we show that this is probably the incorrect explanation given the MAC of brown carbon is actually higher when markers (O:C, toluene:benzene) suggest more oxidation. We suggest a new BrC parameterization is needed to represent wildfire optical properties in the Western United States. We also note that there needs to be better terminology to distinguish between decreasing absorption caused by losses of organic aerosol mass versus decreasing absorption caused by changes in the mass absorption cross section (MAC) of the aerosol. Finally, these results are based on the plume integration method, which might neglect aerosol decay at the edge of the plume where oxidation and evaporation are more rapid compared to the center of the plume. While this effect may be important for studying mechanisms of smoke evolution, it does not affect mean properties, which are what ultimately affect climate and are comparable to modeling results.

**Data Availability**

The WE-CAN data can be found at http://data.eol.ucar.edu/master_lists/generated/we-can/.

The DOI for each data set used in this work are:

PAS and CAPS $PM_{SSA}$: https://doi.org/10.26023/K8P0-X4T3-TN06

PILS1: https://doi.org/10.26023/9H07-MD9K-430D and https://doi.org/10.26023/CRHY-NDT9-C30V

PILS2: https://doi.org/10.26023/7TAN-TZMD-680Y

SP2: https://doi.org/10.26023/P8R2-RAB6-N814
UHSAS: https://doi.org/10.26023/BZ4F-EAC4-290W
PTR-ToF-MS: https://doi.org/10.26023/K9F4-2CNH-EQ0W
HR-AMS: https://doi.org/10.26023/MM2Y-ZGFQ-RB0B
Picarro: https://doi.org/10.26023/NNYM-Z18J-PX0Q
miniQCL: https://doi.org/10.26023/Q888-WZRD-B70F

## Author Contributions

SMM designed the project. YS wrote the paper. YS, RPP, APS, EJTL, LAG, DKF, WP, LH, DWT, TC, EVF, and SMM collected and analyzed data.

## Competing Interests

The authors declare that they have no conflict of interest.

## Acknowledgements

The 2018 WE-CAN field campaign was supported by the U.S. National Science Foundation through grants AGS-1650493 (U of Wyoming), AGS-1650786 (Colorado State U), AGS-1650275 (U of Montana), AGS-1650288 (U of Colorado at Boulder), and the National Oceanic and Atmospheric Administration (Award # NA17OAR4310010, Colorado State U). This material is based upon study supported by the National Center for Atmospheric Research, which is a major facility sponsored by the National Science Foundation under Cooperative Agreement no. 1852977.

The authors acknowledge support from AGS-1650493 for YS, SMM and RPP, AGS-1650786 for APS and EJTL, AGS-2144896 for LH and WP, AGS-1650288 for DWT, NOAA Climate Program Office's Atmospheric Chemistry, Carbon Cycle, and Climate program (Grant NA17OAR4310010) for DKF and LAG.

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
