# Peer review of "Understanding Absorption by Black Versus Brown Carbon in Biomass Burning Plumes from the WE-CAN Campaign"

_EGUsphere, 2023_

## Author Comment (AC1)

**Overall Author Response To Reviewer 1**
We greatly appreciate Reviewer 1's comments. We strongly agree that the "dataset presented in the manuscript is unique and valuable" and we also agree with the idea that the paper was poorly presented and lacked the clarity and "deeper analysis" needed to bring fresh insight to the field. We have dramatically reorganized the paper to clearly emphasize the key findings, which are now clearly described in the abstract, results, conclusions, and detailed below. We have either removed, or relegated to the SI, all the plots with poor correlation coefficients or "unclear" relationships. In our original draft we had elected to show the lack of correlation with variables that are often postulated in the literature as related to brown carbon absorption, but in the revised version we simply say there is no correlation and move any plots to the SI. What remains in the main text is a clear presentation of the key findings, which we strongly believe are significant and move the field forward. We agree with the reviewer that there are difficult, "challenges in drawing definitive conclusions due to the variability in fuels, combustion conditions, and weather among different fires." However, we have not chosen to focus on single transects or single plumes as this type of analysis rarely moves climate or chemical transport models, which cannot have extremely detailed chemistry in them, forward. These are the type of models that must get the optical properties of biomass burning correct, to correctly determine the regional and global radiative forcing of this source. Instead, we now focus the paper on the several characteristics, detailed below, that are consistent across all the fires presented with reasonably strong correlation coefficients.

SUMMARY OF KEY POINTS THAT ARE NOW EMPHASIZED IN THE MANUSCRIPT

1.) Observed mass absorption cross sections (MAC) for brown carbon are significantly lower than common model parameterizations (Saleh et al. parameterization) at both 405 and 660 nm.

2.) We observe a strong relationship between organic aerosol mass (corrected for dilution, OA:CO) and both the organic aerosol O:C ratio ($R^2$=0.8) and the toluene:benzene ratio ($R^2$=0.64), showing that OA is lost with these markers of chemical aging and oxidation. This relationship is only clearly seen when data from all plumes is analyzed together, validating our approach of not focusing on a single plume.

3.) The mass absorption cross section (MAC) of the brown carbon shows no change with these same markers of chemical age and oxidation (O:C, toluene:benzene). If there is any observable trend in MAC, it is an increase with chemical age. However, there is a clear decrease in total absorption at 405 nm and a more subtle decrease in absorption at 660 nm with these markers of chemical age. We conclude that the observed "bleaching" is a decrease in organic mass, not a decrease in MAC or imaginary refractive index. Decreasing imaginary refractive index is currently implemented in several chemical transport and global climate models.

4.) While absorption from brown carbon is much weaker at 660 nm than at 405 nm, it is still significant, representing roughly a quarter of the total absorption at 660 nm. To our knowledge, this is the first time that PILS data has been used to quantify brown carbon absorption at 660 nm.

In conclusion, we have made the manuscript dramatically more straightforward, have clearly explained all of our terminology, and have made important conclusions. We believe the manuscript is now very suitable for publication in ACP.

**Response to Specific Comments from Reviewer 1**

**L62 If BC core is coated by absorbing component, I think the absorption of BC would be reduced.**
If BC core is coated by absorbing component, the absorption enhancements for a BC core and shell with the absorption contribution of the absorbing shell removed would be smaller than the absorption

enhancements for a BC core coated by pure scattering shell, but still larger than 1 (Lack et al., 2010)[1]. That means the absorption of BC core would still be enhanced by the absorbing shell.

**L94 It is unclear in the sentence here. I am not sure if authors want to express BC absorption or BC core absorption is rare. Additionally, MAC BC is for BC core only or the overall BC particle.**
Lines 83-85 now read as "Unfortunately, the MAC of the overall BC particle, $MAC_{BC}$, in the ambient atmosphere continues to be poorly understood due to a lack of field measurements and limitations of filter-based instruments to measure this parameter."

**L139 PM2.5 or PM1**
This is a $PM_{2.5}$ cyclone, which has a $PM_{2.5}$ cut under a flow rate of 3 LPM, and provides a $PM_{1.0}$ cut under a total flow rate of 5.7 LPM. Lines 143-144 now read as "A 3 LPM $PM_{2.5}$ cyclone (URG-2000-30ED) was used on the PAS in front of the inlet to provide a $PM_{1.0}$ cut under a total flow rate of 5.7 LPM."

**L157 I think you should move this paragraph before L152.**
This paragraph has been rewritten and moved to section 2.1. Lines 120-126 now read as "The following instruments are a subset of those flown during the WE-CAN campaign and are utilized in this work. The full WE-CAN dataset is archived at https://data.eol.ucar.edu/master_lists/generated/we-can. All aerosol instruments utilized in this paper, except the PILS, pulled air from the same Solid Diffuser Inlet (SDI) inlet. The PILS sampled from a Submicron Aerosol Inlet (SMAI) (Craig et al., 2013a, 2013b, 2014; Moharreri et al., 2014). All the measurements were converted to standard temperature and pressure (STP, 1 atm, 0ºC) based on the measured temperature and pressure (Eq. 1) before data were uploaded.

$$Variables_{STP} = Variables_{measured} \cdot \frac{Pressure_{STP}}{Pressure_{measured}} \cdot \frac{Temperature_{measured}}{Temperature_{STP}} \qquad (Eq.\,1)$$
"

**L190,L199 Please specify the size range of particles you measured.**
We calculated the average volume mean diameter for each plume using UHSAS, which has been updated to section 2.1.5. Lines 202-204 now reads as "The UHSAS was calibrated with ammonium sulfate. The particle mass concentration was calculated by applying these size bins and multiplying by a particle density of 1.4 g cm$^{-3}$ (Sullivan et al., 2022). The volume mean diameter of the particles for all the detected plumes range between 0.18 mm and 0.34 mm."

**Line 358-360 From Fig 3c and 3d, it is hard to conclude the trend of the fitting line with R2 equals to 0.02 or 0.03**
The comparison with $MAC_{BC660}$ is moved to supplemental, and the rest subplots are removed from the updated manuscript. Lines 571-573 now reads as "$MAC_{BC660}$ is also compared with the physical age and MCE (Fig. S13), the O:C and toluene:benzene chemical clocks (Fig. S14), and the altitude, temperature and dilution (ΔCO) (Fig. S15). However, no clear trend is be found in these comparisons."

**Line 368-370 What is the trend between t/b with O:C ratio? Do they have a nice correlation?**
Toluene:benzene ratio and O:C ratio correlate well with each other. This plot has been added to the updated manuscript, and Lines 398-402 now reads as "The O:C ratio characterizes the oxidation state of OA and typically increases with photochemical age (Aiken et al., 2008), while the toluene:benzene ratio decreases with photochemical processing time since toluene is more reactive than benzene (Gouw et al., 2005). Both markers are two commonly used markers to indicate the chemical age of smoke, and they correlated well with each other during WE-CAN (Fig. S1)."

[Figure]

**Figure S1: Plume integrated toluene:benzene ratio variations with O:C ratio.**

**Line 473 I would suggest you move the part describing how you convert liquid abs to abs in air to the method section.**
This part has been moved to the method section as suggested. Now this part is under section 2.5 (Lines 282-319).

[1]Lack, D. A., & Cappa, C. D. (2010). Impact of brown and clear carbon on light absorption enhancement, single scatter albedo and absorption wavelength dependence of black carbon. *Atmospheric Chemistry and Physics*, 10(9), 4207–4220. https://doi.org/10.5194/acp-10-4207-2010

**Overall Author Response To Reviewer 2**

We greatly appreciate Reviewer 2's comments. As mentioned in the response to Reviewer 1, we have taken the reviewer comments to heart and have made significant changes to the manuscript. We have dramatically reorganized the paper to clearly emphasize the key findings, which are now clearly described in the abstract, results, conclusions, and detailed below. We have either removed, or relegated to the SI, all the plots with poor correlation coefficients and tenuous relationships. In our original draft we had elected to show the lack of correlation with variables that are often postulated in the literature as related to brown carbon absorption, but in the revised version we simply say there is no correlation and move any plots to the SI that have very small correlation coefficients. What remains in the main text is a clear presentation of the key findings, which we strongly believe are significant and move the field forward.

While we agree with most of the reviewer comments, we do disagree that the manuscript suffers from analyzing data from all the wildfires together. While it is indeed difficult to find consistent trends in such a diverse dataset that includes differences in fuel type, fire area, fire maturity, and meteorology, this is exactly what chemical transport and climate models must do to accurately represent the average properties in a model grid cell. For optical properties, it is the results of these types of models that matter in terms of prediction of regional and global radiative forcing from biomass burning aerosol plumes. It is exactly these types of broad relationships that remain true across all fires that our group has used in previous publications (Brown et al., 2021)[2] to assess the accuracy of global climate models. This is now clearly explained in the manuscript, namely that our goal is to provide insights that can be used to understand the accuracy of model predictions. Additionally, even "individual" fire plumes during WE-CAN, were from large fires where a variety of burn conditions and fuel types were already present, meaning even analyzing individual plumes does not remove all of the above-stated variables.

Fortunately, we are able to identify several clear trends related to brown carbon optical properties across different wavelengths, organic mass, and bleaching that have much stronger correlation coefficients than many of the plots previously presented, even given the wide range of fires considered. These are detailed below:

SUMMARY OF KEY POINTS THAT ARE NOW EMPHASIZED IN THE MANUSCRIPT

1.) Observed mass absorption cross sections (MAC) for brown carbon are significantly lower than common model parameterizations (Saleh et al. parameterization) at both 405 and 660 nm.

2.) We observe a strong relationship between organic aerosol mass (corrected for dilution, OA:CO) and both the organic aerosol O:C ratio ($R^2$=0.8) and the toluene:benzene ratio ($R^2$=0.64), showing that OA is lost with these markers of oxidation and chemical aging. This relationship is only clearly seen when data from all plumes is analyzed together, validating our approach of not focusing on a single plume or fire.

3.) The mass absorption cross section (MAC) of brown carbon shows no change with these same markers of oxidation and chemical age (O:C, toluene:benzene). If there is any observable trend in MAC, it is an increase with chemical age. However, there is a clear decrease in total absorption at 405 nm and a more subtle decrease in absorption at 660 nm with these markers of chemical age. We conclude that the observed "bleaching" is a decrease in organic mass, not a decrease in MAC or imaginary refractive index. Decreasing imaginary refractive index is currently implemented in several chemical transport and global climate models.

4.) While absorption from brown carbon is much weaker at 660 nm than at 405 nm, it is still significant, representing roughly a quarter of the total absorption at 660 nm. To our knowledge, this is the first time that PILS data has been used to quantify brown carbon absorption at 660 nm.

[2]Brown, H., Liu, X., Pokhrel, R. *et al.* Biomass burning aerosols in most climate models are too absorbing. *Nat Commun* **12**, 277 (2021). https://doi.org/10.1038/s41467-020-20482-9

**Response to Specific Comments from Reviewer 2**

**The fires themselves are not discussed at all. Analysis of a plume is necessarily incomplete without information on fire location, size, maturity, fuel type, and meteorology. This additional context may provide insight to the various parameters observed.**
Lindaas et al. (2021) has been cited for the information in terms of fire properties in Section 2. Lines 114-118 now reads as "The WE-CAN field campaign consisted of 19 research flights that took place from Jul. 24 – Sep. 13, 2018. Data from 13 flights where all required instrumentation was available were analyzed in this study. The flight path and dominant wildfire for each of these flights are shown in Fig. 1. The fire locations, fuel types for each fire during WE-CAN were characterized and summarized by Lindaas et al. (2021)."
In addition, we didn't separate fires based on the fuel types and burn conditions, because we want to provide general conclusions that can be applied to improve models. This reason is now explained and emphasized in Section 3.2 (Lines 384-388), which reads, "Because all large wildfire emissions are a mix of different regions that are burning slightly different fuels at different combustion efficiencies and because models treat regions, not individual fires, we identify relationships in this paper that hold true across all the flight data collected during WE-CAN. These types of broad correlations are much more useful than individual case studies yielding results that only hold true sometimes."

**Line 152: How were the optical parameters translated to STP? Which equations were used?**
We used the ideal gas law, the ambient temperature and pressure measured by the instruments to convert the measurements to STP and the equation has been included in Section 2.1. Lines 120-126 now read as "The following instruments are a subset of those flown during the WE-CAN campaign and are utilized in this work. The full WE-CAN dataset is archived at https://data.eol.ucar.edu/master_lists/generated/we-can. All aerosol instruments utilized in this paper, except the PILS, pulled air from the same Solid Diffuser Inlet (SDI) inlet. The PILS sampled from a Submicron Aerosol Inlet (SMAI) (Craig et al., 2013a, 2013b, 2014; Moharreri et al., 2014). All the measurements were converted to standard temperature and pressure (STP, 1 atm, 0ºC) based on the measured temperature and pressure (Eq. 1) before data were uploaded.

$$Variables_{STP} = Variables_{measured} \cdot \frac{Pressure_{STP}}{Pressure_{measured}} \cdot \frac{Temperature_{measured}}{Temperature_{STP}} \qquad (Eq. 1)$$
"

**Line 197: Same question: how were SP2 measurements converted to STP?**
We used the same equation with the optical variables' conversion, and the equation has been included in Section 2.1 (Lines 120-126).

**Section 2.2: Why did the authors not use a more robust method for calculating plume age, such as HYSPLIT?**
We think the plume age calculated from the distance dividing by the wind speed is more accurate than the HYSPLIT model. Because the actual distance from the center of the plume transect to the burn area and the measured average wind speed across the transect were used in the calculation. Also, this methodology for plume age measurement is consistent with other published manuscripts from WE-CAN as is now stated in the Section 2.2. Line 226-231 now read as "The physical age of the plume was calculated by dividing the distance the plume was sampled from the fire source by the in-plume average wind speed. The average

wind speed was measured on the NSF/NCAR C-130 aircraft during each plume pass. The distance was estimated by using the longitude and latitude of the geometric center of the plume measured on the NSF/NCAR C-130 and the fire location provided by the U.S. Forest Service. The same method was used by Garofalo et al. (2019), Peng et al. (2020), Lindaas et al. (2021), Permar et al. (2021), and Sullivan et al. (2022) and are also utilized here for consistency."

**Line 337: The authors state "This result indicates that the combustion conditions (flaming or smoldering) does not have an easily described relationship to $MAC_{BC660}$." I believe this result is mostly indicative of how difficult it is to use MCE to describe a fire. MCE is highly variable and may have vastly different values at different locations within the fire area. The authors may want to consider using another plume marker - the PTR-ToF-MS may provide a measurement of HCN, though this will likely carry similar uncertainty.**
We agree that MCE is difficult to use to predict the properties of fire emissions. We do not focus on MCE in the manuscript. Analysis of the relationship to HCN may be interesting, but is beyond the scope of the current publication, which now focuses on the robust conclusions stated above.

**Line 357: The statement "Even for each individual flight, the increasing trend in mean diameter is clear" is not supported by Figure S2.**
We agree that this overstates the relationship. This statement and this plot are removed from the updated manuscript.

**Line 358: The statement "Another contributor to increasing SSA is the decrease in absorption at 660 nm (Fig. 3d) with age for most fires" is not supported by figure 3d. Perhaps overall, the linear regression shows a negative slope, but when considering individual plumes (notably RF10 and RF19), the interpretation fails.**
The comparison with $MAC_{BC660}$ is moved to supplemental, and the other subplots of Figure 3 have been removed from the updated manuscript. The text on Lines 571-573 now reads as "$MAC_{BC660}$ is also compared with the physical age and MCE (Fig. S13), the O:C and toluene:benzene chemical clocks (Fig. S14), and the altitude, temperature and dilution ($\Delta CO$) (Fig. S15). However, no clear trend is be found in these comparisons."

**The authors do not justify the use of linear regressions as their model fit for these data and it is unlikely that any of the several parameters would be linear in the others, save, perhaps OA/CO vs O:C Ratio, which is rather obvious. Most of the regressions in this analysis show no linear correlation whatsoever, yet the authors discuss the results as if the correlations are significant (line 408, for example). The authors may want to consider different models when fitting their reanalyzed data.**
All the plots without reasonably large correlation coefficients have been removed or moved to supplemental. Those that remain with low correlation coefficients are used to demonstrate that there isn't a clear correlation. We now use ODR fits instead of the least square fits. Linear fits do reasonably well for the plots and correlations that we now emphasize in the manuscript. We cannot justify a different type of fit given the difference in correlation coefficients for other fits vs. the simple linear model. We disagree with the reviewer that OA/CO vs. O:C Ratio is a "rather obvious" correlation. In fact, there is a large amount of disagreement in the literature if organic mass increases or decreases with oxidation/chemical aging, which occurs at the same time the plumes are diluting, in plumes from biomass burning. We believe this is a key conclusion that significantly moves the field forward. It is indeed not at all obvious that oxidation would add more mass than is removed by dilution. OA/CO is an extensive property that gives the dilution corrected total mass of organic aerosol, while O:C ratio is an intensive property related to oxidation level of the aerosol.

**In Figure 7, there is clearly no correlation between either altitude or temperature and OA/CO, rather, these graphs taken together merely show that colder plumes are found at higher altitudes. The same can be said of Figure 9 (a) and (b).**
These plots have been moved to the supplemental. We agree that there is no clear trend, but leave the plots in the SI because it is perhaps interesting that all the low altitude/high temperature plumes seem to have less organic mass, though we don't have a broad enough dataset to make a robust conclusion. The related text on Lines 490-494 now reads as "Plume-integrated CO-normalized OA also shows weak or no trend with altitude and temperature (Fig. S5). However, we note that the smallest OA:CO was captured in the plumes (RF08) that have highest temperature (~305 K), and larger OA:CO tends to be observed in the colder plumes (RF19). More studies are needed to determine how much OA is evaporated in high temperature plumes because the WE-CAN dataset does not capture enough variation of temperature within plumes to make a robust conclusion.", and text on Lines 571-573 now reads as "$MAC_{BC660}$ is also compared with the physical age and MCE (Fig. S13), the O:C and toluene:benzene chemical clocks (Fig. S14), and the altitude, temperature and dilution ($\Delta CO$) (Fig. S15). However, no clear trend is be found in these comparisons."

**Line 459: Are the authors trying to draw a distinction between externally and internally mixed BrC?**
We are distinguishing absorption at 660 nm from the lensing effect, and BrC. These parameters do not give a direct measure of if the BrC is externally or internally mixed. Determination of the mixing state of the aerosol is beyond the scope of this manuscript other than to say that the fact there is significant lensing observed suggests at least some of the aerosol is internally mixed.

**Line 470: please cite the Mie code used, unless it was developed by the authors. If so, please state so.**
The Mie code is based on Bohren and Huffman (1983) and the author who implemented Mie code into Igor code is now thanked in the acknowledgments (Line 671). Lines 293-298 now read as "Here we use Mie theory (Bohren and Huffman, 1983) to convert absorption from BrC in aqueous solution to the absorption from BrC particles in the ambient (Liu et al., 2013; Zeng et al., 2020). The complex refractive index ($m = n + ik$) was put into a Mie code (implemented into Igor by Ernie R. Lewis base on Bohren and Huffman, 1983) to obtain the absorption efficiency (Q), and further used to calculate the absorption coefficient by Eq. 7 (Liu et al., 2013). The real part of the refractive index (n) is set to be 1.55, and the imaginary part is calculated by using Eq. 8 (Liu et al., 2013)."

**Line 561: Similar to other figures where the r2 is extremely low, there are no trends in Figure 14 and no conclusions can be drawn from the data as presented.**
We agree. Again, all the plots without obvious correlations have been removed or moved to supplemental, unless the plot is included to show there isn't a correlation where one might be expected based on past literature. The related text on Lines 410-420 now reads as "Very weak or non-trends are observed versus the chemical markers of aging (Fig. 3). If there is any trend, it is a slight increase in $MAC_{ws\_BrC405}$ with O:C ratio with a poor correlation. A similar weak trend is also observed when compared $MAC_{ws\_BrC405}$ and $MAC_{BrC+lensing\_405}$ with the toluene:benzene ratio (Fig. S2). The flat or slightly increasing trend with increasing oxidation level and decreasing toluene:benzene suggests that the refractive index of BrC is not changing in a consistent way at 405 nm. It is important to remember that most of the trends observed in WE-CAN are caused by emissions from different fires versus variations within a single fire, which tend to be quite small. Only 2 flights shows a clear trend ($R^2 > 0.3$) for both $MAC_{ws\_BrC405}$ and $MAC_{BrC+lensing\_405}$ with increasing O:C ratio at the same time, and they are RF03 ($R^2$ of 0.85 and 0.85 with positive slope for $MAC_{ws\_BrC405}$ and $MAC_{BrC+lensing\_405}$), and RF06 ($R^2$ of 0.8 and 0.49 with negative slope for $MAC_{ws\_BrC405}$ and $MAC_{BrC+lensing\_405}$), where RF03 only measured a single fire (Taylor Creek fire)."

---

## Author Response (AR2)

**Response to Specific Comments from Reviewer 1**

**Line 43 While I agree that much of the terminology in this field is operationally defined, the terms EC, soot, and others listed here overlap. This sentence needs correction and rephrasing.**
Line 43-46 now reads as "BC is somewhat poorly defined, but is generally considered to be insoluble, refractory, and has an absorption exponent near one. Other materials such as elemental carbon (EC), and soot (Wei et al., 2013) are often very similar to BC, but each is operationally defined by how it is measured."

**Line 77 As the authors mentioned the mixing state and absorption enhancement in this paragraph, more details on BC heterogeneity, which mitigates the theoretical enhancement, should be discussed here.**
Discussion about the absorption enhancement caused by changes in BC morphology and absorbing shell versus pure scattering shell are now added into this paragraph as suggested. Line 74-83 now reads as "Observations of absorption enhancement from ambient BC vary widely in field studies due to variations in coating thickness, coating material, source type, or methodological differences, but it is often much lower than laboratory values (Liu et al., 2015, 2017; Cappa et al., 2012, 2019; Healy et al., 2015; Krasowsky et al., 2016). Cappa et al. (2019) summarized absorption enhancements observed at the red end of the visible spectrum from 10 studies including ambient measurements, source sampling, and lab experiments. The absorption enhancement reported by those measurements ranged from 1.1 to 2.8. Lack et al. (2010) found that the absorption enhancement caused by the absorbing shell would be smaller than the absorption enhancement caused by the pure scattering shell. The non-spherical morphology of BC and the tendency of BC to compact when coated by organics also can both enhance and decrease absorption (Romshoo et al., 2021; Kelesidis et al., 2022)."

Romshoo, B., Müller, T., Pfeifer, S., Saturno, J., Nowak, A., Ciupek, K., Quincey, P. and Wiedensohler, A.: Optical properties of coated black carbon aggregates: Numerical simulations, radiative forcing estimates, and size-resolved parameterization scheme, Atmos. Chem. Phys., 21(17), 12989–13010, doi:10.5194/acp-21-12989-2021, 2021.

Kelesidis, G. A., Neubauer, D., Fan, L. S., Lohmann, U. and Pratsinis, S. E.: Enhanced Light Absorption and Radiative Forcing by Black Carbon Agglomerates, Environ. Sci. Technol., 56(12), 8610–8618, doi:10.1021/acs.est.2c00428, 2022.

**Line 79 The expression is not professional enough and is too colloquial. This issue is prevalent throughout the manuscript.**
Line 84 now reads as "The mass absorption cross section of BC ($MAC_{BC}$) is an alternative method to quantify the absorbing ability of BC containing particles versus absorption enhancement." While we have changed this particular sentence, we disagree with the reviewer's comment that the manuscript overall is too colloquial and not professional enough. All the co-authors have published extensively and nearly all native speakers of English, who have reviewed and edited this manuscript multiple times. We have reviewed the manuscript yet again and feel there are minimal grammar or phrasing issues that have not been resolved at this point. We strongly believe that scientific papers should, to the extent possible, avoid jargon and complex phrasing in an effort to explain things in a straightforward way for readers to understand.

**Line 81 "At a certain wavelength" might be a better phrasing.**
Line 87 now reads as "$MAC_{BC}$ is the particulate absorption divided by the mass of the pure BC at a certain wavelength" as suggested.

**Line 86 to Line 93 The authors only list results from previous publications without providing their own summary. A synthesis or summary is needed here.**

Line 92-100 now reads as "Bond and Bergstrom (2006) suggested a $MAC_{BC}$ of 7.5±1.2 m$^2$ g$^{-1}$ at 550 nm for fresh BC. The following campaigns demonstrate the variety of $MAC_{BC}$ measured in the ambient during the past 15 years. Subramanian et al. (2010) reported a $MAC_{BC}$ of 10.9±2.1 m$^2$ g$^{-1}$ at 660 nm and 13.1 m$^2$ g$^{-1}$ at 550 nm over Mexico City when using a single particle soot photometer (SP2) and the filter-based particle soot absorption photometer (PSAP) instrument during airborne measurements. Krasowsky et al. (2016) reported a $MAC_{BC}$ enhancement of 1.03±0.05 due to the coatings on BC. Zhang et al. (2017) found a $MAC_{BC}$ with a mean of 10 m$^2$ g$^{-1}$ and a standard deviation of 4 m$^2$ g$^{-1}$ at 660 nm by using both SP2 and PSAP measurements. Cho et al. (2019) summarized $MAC_{BC}$ estimated from more than 10 studies in East and South Asia in both ambient conditions and laboratory experiments, and the values ranged from 4.6 to 11.3 m$^2$ g$^{-1}$."

**Line 126 There is no need to include this basic equation in the main manuscript.**
This equation was requested by the second reviewer. We think this equation might also be helpful to people who don't have experience on airborne measurements.

**Line249 In Equation 2, Eabs includes the effects of both lensing and BrC. If using a wavelength of 660 nm, BrC might be negligible. The same applies to Equation 3. The wavelength used should be specified here, rather than just using lambda.**
Thank you for pointing out this. We only compared $E_{abs}$ and $MAC_{BC}$ at 660 nm in this work. Line 257-268 now reads as "Absorption enhancement ($E_{abs}$) is the ratio of the absorption of all particles (including BC core and coating materials) to the absorption of BC alone (Lack and Cappa, 2010). $E_{abs}$ at 660 nm ($E_{abs\_660}$) was calculated in this study by Eq. 2:

$$E_{abs\_660} = \frac{Abs_{Total\_660}}{M_{BC} * MAC_{BC\_core\_660}} \qquad (Eq.\,2)$$

where $Abs_{Total\_660}$ is the total absorption coefficient at a wavelength of 660 nm measured by the PAS, $M_{BC}$ is the mass concentration of BC measured by the SP2, and $MAC_{BC\_core\_660}$ is the MAC of BC alone (without any other coating material) at 660 nm, which is set to be 6.3 m$^2$ g$^{-1}$ (Bond and Bergstrom, 2006; Subramanian et al., 2010).
$MAC_{BC}$ at 660 nm was calculated following Eq. 3:
$$MAC_{BC\_660} = \frac{Abs_{Total\_660}}{M_{BC}} \qquad (Eq.\,3)"$$

**Line 311 Did Mie factors vary significantly in each plume?**
We only calculate plume integrated Mie factors to avoid noise caused by rapid changes within a plume. The variation of Mie factors within the same plume is beyond the scope of the current publication.

**Line 362 One reason for discrepancies might be that Saleh's study focused on emissions, while samples collected during WE-CAN might have experienced some degree of aging.**
Agreed. Line 371-373 now reads as "The discrepancy could also be partly because the data Saleh et al. used for their parameterization comes from controlled laboratory burns and not wildfires or because emissions observed during WE-CAN have all undergone some near-source aging before being observed by the aircraft."

**Line 376 The change in the absorption coefficient is due to changes in RI, MAC (intrinsic properties), and mass. Since the mass concentration of chromophores is unknown, most studies normalize the absorption coefficient to the OA concentration to obtain the MAC value. The basic assumption that all OA absorb equally need to specify in the context.**

Line 387-392 now reads as "While only a fraction of organic aerosol mass absorbs light, both models and observations typically treat all organics as equally absorbing because of the inability to distinguish absorbing and non-absorbing molecules. The definition of bleaching or whitening was unclear in previous literature. Models tend to treat bleaching as the change of refractive index or decreasing of MAC (Brown et al., 2018; Wang et al., 2018; Carter et al., 2021), while observations or lab experiments mostly link bleaching to the decrease of normalized total absorption (Forrister et al., 2015; Palm et al., 2020; Zeng et al., 2022)." This is also specified in section 2.4 after equations. Line 267-279 now reads as "MAC of BrC and lensing is calculated at 405 and 660 nm (Eq. 4):

$$MAC_{BrC+lensing\_\lambda} = \frac{Abs_{Total\_\lambda} - M_{BC} * MAC_{BC\_core\_\lambda}}{M_{OA}} \qquad (Eq.4)$$

where $M_{OA}$ is the organic mass measured by the AMS. Again, the $MAC_{BC\_core\_\lambda}$ is set to be 6.3 and 10.2 m$^2$ g$^{-1}$, respectively, at 660 nm and 405 nm yielding an absorption Ångström exponent (AAE, the negative slope of a logarithmic absorption coefficient against wavelength) of 0.99 for the BC core (Bond and Bergstrom, 2006; Subramanian et al., 2010; Liu, et al., 2015). It should be noted that both BrC and lensing contribute to the MAC$_{BrC+lensing\_\lambda}$, and cannot be separated using this approach and MAC$_{BrC+lensing\_\lambda}$ is the MAC of all organics without distinguishing absorbing and non-absorbing particles.

MAC of water-soluble BrC at $\lambda$ nm (MAC$_{ws\_BrC\lambda}$) is calculated using Eq. 5:

$$MAC_{ws\_BrC\lambda} = \frac{Abs_{ws\_BrC\_\lambda}}{WSOC * (WSOM:WSOC)} \qquad (Eq.5)$$

where $Abs_{ws\_BrC\_660}$ is water-soluble light absorption and WSOC is water-soluble organic carbon mass, which are both measured by the PILS system. WSOM:WSOC ratio is set to be 1.6 (Sullivan et al., 2022). MAC$_{ws\_BrC\lambda}$ is the MAC of all water-soluble organics without distinguishing absorbing and non-absorbing particles."

**Line 440 How do the authors define a near-source flight?**
We are referring to the plumes where we directly followed the emissions from as near to the flame front as we were allowed to go. Line 452-454 now reads as "Despite the much longer transit time and distance, overall these emissions, which were measured 300 to 600 miles away, have a very similar MAC$_{BrC+lensing\_405}$ to that of the near-source flights where we tracked emissions from as near to the fire source as allowed by air traffic control."

**Line 445 I suggest the authors examine BC with CO to align the data better.**
We aligned all the measurement with CO, but the SP2 data sometimes would show a hysteresis due to the manually dilution, which is to prevent signal saturation. This is not an issue when using plume integrated method, because the whole peak of SP2 will be integrated. However, the plume is highly mixed during RF05 and RF08, and the plume integration method is not applicable anymore. Time averaging is applied for these two flights and can not fully removing the extreme values caused by the SP2 hysteresis. Line 458-460 now reads as "There are several extreme values that exist in the dataset, probably because of the SP2 hysteresis caused by variation in the dilution rate of the SP2 which cannot be totally eliminated from the 1-minute average."

**Line 458 Have the authors accounted for the Mie factor? I suggest using different symbols to represent absorption in liquid or in air.**
Abs$_{BrC+lensing\_405}$ is the absorption coefficient of water-soluble BrC at 405 nm, which is directly measured by PILS, and does not account for the Mie factor. Line 472-474 now reads as "The average water-soluble BrC absorption at 405 nm (Abs$_{ws\_BrC405}$, 0.02 Mm-1 ppbv-1) which is directly measured by the PILS, is only 20% of the total absorption from BrC plus lensing (Abs$_{BrC+lensing\_405}$, 0.11 Mm-1 ppbv-1), which is calculated from the PAS and SP2 (Eq.9).

**Line 506 Absorption at 660 nm could also be due to the penetration of BC particles into the filter. A 0.2 μm filter is not sufficient to filter out all BC particles. How can the authors rule out the presence of BC?**

According to Peltier et al. (2007), black carbon adheres to the PILS impactor and is not efficiently transferred to the liquid stream. The inline 0.2 μm liquid filter also helps remove more. They also pointed out that black carbon is also not oxidized by the TOC Analyzer, and any insoluble particles greater than 110 nm could not be oxidized by the TOC Analyzer based on tests with PSL. Line 521-523 now reads as "We know that absorption observed in the PILS at 660 nm is not BC because BC is insoluble and will be removed by the PILS impactor, the 0.2 μm filter in the instrument, and that BC over 110 nm in size will not be oxidized by the TOC analyzer (Peltier et al., 2007; Zeng et al., 2021; Sullivan et al., 2022)."

**Response to Specific Comments from Reviewer 2**

**In section 2.1, the authors do not justify (with literature citations or otherwise) the conversion of optical properties to STP. I have never seen this done before, which does not mean it is an incorrect procedure, but the authors should add detail about what errors in (specifically) optical properties would arise if the data were not corrected, and cite previous literature showing the background and scientific basis of this procedure. Note that this is different than the pressure-dependent correction to the photoacoustic absorption signal.**

The instruments on the aircraft sampled air though different inlets that had different residence times and thus the measured air arrived at the instruments at different temperatures and pressures. To make instruments comparable, the measurement must be converted to either ambient conditions or STP. STP conversion was chosen during WE-CAN for two reasons 1.) It introduces less error because only the measured temperature and pressure at the instrument location is used instead of also needing the measured temperature and pressure of the ambient air and 2.) it eliminates artifacts that would arise from air being measured at different altitudes having the same mixing ratios, but different mass concentrations. Mixing ratios are typically used for the gas-phase species (ppb etc.) and these are constant with variations in pressure if the amount of species remains constant. However, aerosol properties are typically reported as densities or concentrations ($ug/m^3$, etc.) because we don't know the molecular weight of the aerosol and cannot easily generate a mixing ratio. The concentrations will change with pressure even if the mixing ratios do not, therefore it is best practice to convert all concentrations to STP for airborne data to avoid a situation where mixing ratios remain constant with pressure while concentrations change even though the relative amounts of the species are constant. This is especially true for our study where we are interested in chemical changes causing changes in the ratio or aerosol to gas and not changes that would occur purely because of pressure changes. Optical properties (mass absorption coefficient, etc.) also need to be adjusted in the same way to be consistent, otherwise they would also vary with pressure. As noted in the paper, the entire WE-CAN dataset (https://data.eol.ucar.edu/master_lists/generated/we-can) has been converted to STP for these reasons and this data has been used in numerous other papers from WE-CAN including Palm et al., 2020 and Sullivan et al., 2022, which are already cited in the manuscript. The errors introduced by this conversion, which are small because we can accurately measure both temperature and pressure in the inlet lines, were included in the posted data.

Palm, B. B., Peng, Q., Fredrickson, C. D., Lee, B. H., Garofalo, L. A., Pothier, M. A., Kreidenweis, S. M., Farmer, D. K., Pokhrel, R. P., Shen, Y., Murphy, S. M., Permar, W., Hu, L., Campos, T. L., Hall, S. R., Ullmann, K., Zhang, X., Flocke, F., Fischer, E. V. and Thornton, J. A.: Quantification of organic aerosol and brown carbon evolution in fresh wildfire plumes, Proc. Natl. Acad. Sci. U. S. A., 117(47), 29469–29477, doi:10.1073/pnas.2012218117, 2020.

Sullivan, A. P., Pokhrel, R. P., Shen, Y., Murphy, S. M., Toohey, D. W., Campos, T., Lindaas, J., Fischer, E. V. and Collett, J. L.: Examination of Brown Carbon Absorption from Wildfires in the Western U.S. During the WE-CAN Study, Atmos. Chem. Phys. Discuss., (July), 1–29, doi:10.5194/acp-2022-459, 2022.

---

## Author Response (AR3)

Author's response for

# Understanding the Mechanism and Importance of Brown

# Carbon Bleaching Across the Visible Spectrum in Biomass

# Burning Plumes from the WE-CAN Campaign

Yingjie Shen[1], Rudra P. Pokhrel[1, a], Amy P. Sullivan[2], Ezra J. T. Levin[2, b], Lauren A. Garofalo[3],
Delphine K. Farmer[3], Wade Permar[4], Lu Hu[4], Darin W. Toohey[5], Teresa Campos[6], Emily V.
Fischer[2], Shane M. Murphy[1]

[1]Department of Atmospheric Science, University of Wyoming, Laramie, WY 82071, USA.
[2]Department of Atmospheric Science, Colorado State University, Fort Collins, CO 80523, USA
[3]Department of Chemistry, Colorado State University, Fort Collins, CO 80523, USA
[4]Department of Chemistry and Biochemistry, University of Montana, Missoula, MT 59812, USA.
[5]Department of Atmospheric and Oceanic Sciences, University of Colorado Boulder, Boulder, CO 80309, USA
[6]National Center for Atmospheric Research, Atmospheric Chemistry Division, Boulder, CO 80301, USA

[a]now at Air Pollution Control Division, Colorado Department of Public Health and Environment, Denver, CO 80246,
USA
[b]now at METEC Research Group, Colorado State University Energy Institute, Fort Collins, CO 80524, USA

*Corresponding author: Shane M. Murphy (shane.murphy@uwyo.edu)*

We modified Line 130-139 to include a brief justification of the conversion of optical properties to STP as suggested by editor and reviewer 2, and nothing was changed in the supplement document.